# Testing Screened Modified Gravity

**Philippe Brax** [1,*], **Santiago Casas** [2], **Harry Desmond** [3] and **Benjamin Elder** [4]

[1] Institut de Physique Théorique, Université Paris-Saclay, CEA, CNRS, CEDEX, F-91191 Gif-sur-Yvette, France
[2] Institute for Theoretical Particle Physics and Cosmology (TTK), RWTH Aachen University, D-52056 Aachen, Germany; casas@physik.rwth-aachen.de
[3] McWilliams Center for Cosmology, Department of Physics, Carnegie Mellon University, 5000 Forbes Ave, Pittsburgh, PA 15213, USA; hdesmond@andrew.cmu.edu
[4] Department of Physics and Astronomy, University of Hawai'i, 2505 Correa Road, Honolulu, HI 96822, USA; bcelder@hawaii.edu
\* Correspondence: philippe.brax@ipht.fr

**Abstract:** Long range scalar fields with a coupling to matter appear to violate known bounds on gravitation in the solar system and the laboratory. This is evaded thanks to screening mechanisms. In this short review, we shall present the various screening mechanisms from an effective field theory point of view. We then investigate how they can and will be tested in the laboratory and on astrophysical and cosmological scales.

**Keywords:** screening; modified gravity; dark energy





## 1. Introduction: Why Light Scalars?

Light scalar fields are mainly motivated by two unexplained phenomena, the existence of astrophysical effects associated with dark matter [1,2] and the apparent acceleration of the expansion of the Universe [3,4]. In both cases, traditional explanations exist. Dark matter could be a Beyond the Standard Model (BSM) particle (or particles) with weak interactions with ordinary matter (WIMPs) [5]. The acceleration of the Universe could be the result of the pervading presence of a constant energy density, often understood as a pure cosmological constant term [6] in the Lagrangian governing the dynamics of the Universe on large scales, whose origin remains mysterious [7,8]. Lately, this standard scenario, at least on the dark matter side, has been challenged due to the lack of direct evidence in favour of WIMPS at accelerators or in large experiments dedicated to their search (Xenon1T and similar experiments) [9–20]. In this context, the axion or its related cousins, the ALP's (Axion-Like Particles) have come back to the fore [21]. More generally, (pseudo-)scalars could play the role of dark matter thanks to the misalignment mechanism, i.e., they behave as oscillating fields, as long as their mass $m$ is low, typically $m \lesssim 1$ eV [22–24].

On the late acceleration side, the cosmological constant is certainly a strong contender, albeit a very frustrating one. The complete absence of dynamics required by a constant vacuum energy is at odds with what we know about another phase of acceleration, this time in the very early Universe, i.e., inflation [25–28]. This is the leading contender to unravel a host of conundrums, from the apparent isotropy of the Cosmic Microwave Background (CMB) to the generation of primordial fluctuations. The satellite experiment Planck [29] has taught us that the measured non-flatness of the primordial power spectrum of fluctuations is most likely due to a scalar rolling down its effective potential. This and earlier results have prompted decades of research on the possible origin of the late acceleration of the Universe.

In most of these models, scalar fields play a leading role and appear to be very light on cosmological scales, with masses sometimes as low as the Hubble rate now, $10^{-33}$ eV [30,31]. Quantum mechanical considerations and in particular the presence of gravitational interactions always generate interactions between these scalars and matter.

The existence of such couplings is even de rigueur from an effective field theory point of view (in the absence of any symmetry guaranteeing their absence) [32]. Immediately this leads to a theoretical dead end, however, as natural couplings to matter would inevitably imply strong violations of the known bounds on the existence of fifth forces in the solar system, e.g., from the Cassini probe [33]. As a typical example, $f(R)$ models [34] with a normalised coupling to matter of $\beta = 1/\sqrt{6}$ belong to this category of models, which would be excluded if non-linearities did not come to the rescue [35].

These non-linearities lead to the screening mechanisms that we review here. We do so irrespective of the origin and phenomenology of these scalar fields, be it dark matter- or dark energy-related, and present the screening mechanisms as a natural consequence of the use of effective field theory methods to describe the dynamics of scalar fields at all scales in the Universe. Given the ubiquity of new light degrees of freedom in modified gravity models—and the empirical necessity for screening—screened scalars represent one of the most promising avenues for physics beyond $\Lambda$CDM.

There are a number of excellent existing reviews on screening and modified gravity [36–45]. In [36], the emphasis is mostly on chameleons, in particular the inverse power-law model, and symmetrons. K-mouflage is reviewed in [37] together with Galileons as an example of models characterised by the Vainshtein screening mechanism. A very comprehensive review on screening and modified gravity can be found in [39], where the screening mechanisms are classified into non-derivative and derivative, up to second-order, mechanisms for the first time. There are subsequent more specialised reviews such as [42] on the chameleon mechanism, and [43] with an emphasis on laboratory tests. Astrophysical applications and consequences are thoroughly reviewed in [40,41], whilst more theoretical issues related to the construction of scalar-tensor theories of the degenerate type (DHOST) are presented in [45]. Finally, a whole book [44] is dedicated to various approaches to modified gravity. In this review, we present the various screening mechanisms in a synthetic way based on an effective field theory approach. We then review and update results on the main probes of screening from the laboratory to astrophysics and then cosmology with future experiments in mind. Some topics covered here have not been reviewed before and range from neutron quantum bouncers to a comparison between matter spectra of Brans-Dicke and K-mouflage models.

We begin with a theoretical overview (Section 2) before discussing tests of screening on laboratory (Section 3), astrophysical (Section 4) and cosmological (Section 5) scales. Section 6 summarises and discusses the complementarity (and differences) between these classes of tests.

## 2. Screening Light Scalars

### 2.1. Coupling Scalars to Matter

Screening is most easily described using perturbations around a background configuration. The background could be the cosmology of the Universe on large scales or the solar system. The perturbation is provided by a matter overdensity. This could be a planet in the solar system, a test mass in the laboratory or matter fluctuations on cosmological scales. We will simplify the analysis by postulating only a single scalar $\phi$, although the analysis is straightforwardly generalised to multiple scalars. The scalar's background configuration is denoted by $\phi_0$ and the induced perturbation of the scalar field due to the perturbation by an overdensity will be denoted by $\varphi \equiv \phi - \phi_0$. At the lowest order in the perturbation and considering a setting where space-time is locally flat (i.e., Minkowski), the Lagrangian describing the dynamics of the scalar field coupled to matter is simply [37,46]

$$\mathcal{L}_2 = \frac{Z(\phi_0)}{2}(\partial_\mu \varphi)^2 + \frac{m_\phi^2(\phi_0)}{2}\varphi^2 - \delta g_{\mu\nu}\delta T^{\mu\nu} + \dots \qquad (1)$$

at the second-order in the scalar perturbation $\varphi$ and the matter perturbation $\delta T_{\mu\nu}$. The latter is the perturbed energy-momentum tensor of matter compared to the background. In this Lagrangian, matter is minimally coupled to the perturbed Jordan metric $\delta g_{\mu\nu}$ and the

Jordan frame energy-momentum tensor is therefore conserved $\partial_\mu \delta T^{\mu\nu} = 0$. The expansion of the Lagrangian starts at second-order as the background satisfies the equations of motion of the system. Notice that we restrict ourselves to situations where the Lorentz invariance is preserved locally. For instance, in the laboratory, we assume that the time variation of the background field is much slower than the ones of experiments performed on Earth. There are three crucial ingredients in this Lagrangian. The first is $m(\phi_0)$, i.e., the mass of the scalar field. The second is the wave function normalisation $Z(\phi_0)$. The third is the composite metric $\delta g_{\mu\nu}$, which is not the local metric of space-time but the leading 2-tensor mediating the interactions between the scalar field and matter. This Jordan metric can be expanded as

$$\delta g_{\mu\nu} = \frac{\beta(\phi_0)}{m_{\text{Pl}}}\varphi\eta_{\mu\nu} - \gamma(\phi_0)\partial_\mu\partial_\nu\varphi + \delta(\phi_0)\partial_\mu\varphi\partial_\nu\varphi + \dots. \tag{2}$$

At the leading order, the first term is the dominant one and corresponds to a conformal coupling of the scalar to matter with the dimensionless coupling constant $\beta(\phi_0)$[1]. One can also introduce a term in second derivatives of $\varphi$, which depends on a dimensionful coupling of dimension minus three. Finally, going to higher order, there are also terms proportional to the first-order derivatives of $\varphi$ squared and a coupling constant of dimension minus four. These two terms can be seen as disformal interactions [47].

The equations of motion for $\varphi$ are given by

$$\partial_\mu(Z(\phi_0)\partial^\mu\varphi) - m^2(\phi_0)\varphi - 2\partial_\mu(\delta(\phi_0)\partial_\nu\varphi)\delta T^{\mu\nu} = -\frac{\beta(\phi_0)}{m_{\text{Pl}}}\delta T + \partial_\mu\partial_\nu(\gamma(\phi_0))\delta T^{\mu\nu} \tag{3}$$

where $\delta T \equiv \delta T^\mu_\nu$, and we have used the conservation of matter $\partial_\mu\delta T^{\mu\nu} = 0$. This equation will allow us to describe the different screening mechanisms.

### 2.2. Modified Gravity

Let us now specialise the Klein–Gordon equation to experimental or observational cases where $\delta T^{00} = \rho$ is a static matter overdensity locally and the background is static too. This corresponds to a typical experimental situation where overdensities are the test masses of a gravitational experiment. In this case, we can focus on the case where $\phi_0$ can be considered to be locally constant. As a result, we have

$$K^{\mu\nu}(\phi_0)\partial_\mu\partial_\nu\varphi - m^2(\phi_0)\varphi = -\frac{\beta(\phi_0)}{m_{\text{Pl}}}\delta T. \tag{4}$$

The kinetic terms are modified by the tensor

$$K^{\mu\nu}(\phi_0) = Z(\phi_0)\eta^{\mu\nu} - 2\delta(\phi_0)\delta T^{\mu\nu}. \tag{5}$$

When the overdensities are static, the disformal term in $K^{\mu\nu}$, which depends on the matter energy momentum tensor, does not contribute and we have $K^{ij}(\phi_0) \simeq Z(\phi_0)\delta^{ij}$ leading to the modified Yukawa equation

$$\Delta\varphi - \frac{m^2(\phi_0)}{Z(\phi_0)}\varphi = \frac{\hat{\beta}(\phi_0)}{m_{\text{Pl}}}\rho. \tag{6}$$

where $\hat{\beta}(\phi_0) = \frac{\beta(\phi_0)}{Z(\phi_0)}$. For nearly massless fields we can neglect the mass term within the Compton wavelength of size $m^{-1}(\phi_0)$, which is assumed to be much larger than the experimental setting. In this case, the Yukawa equation becomes a Poisson equation

$$\Delta\varphi = \frac{\hat{\beta}(\phi_0)}{m_{\text{Pl}}}\rho. \tag{7}$$

As a result, the scalar field behaves similarly to the Newtonian potential, and the matter interacts with the effective Newtonian potential[2]

$$\Phi = \Phi_N + \frac{\beta(\phi_0)}{m_{\text{Pl}}}\varphi = \left(1 + \frac{2\beta^2(\phi_0)}{Z(\phi_0)}\right)\Phi_N \tag{8}$$

i.e., gravity is modified with an increase in Newton's constant by

$$G_{\text{eff}} = \left(1 + \frac{2\beta^2(\phi_0)}{Z(\phi_0)}\right)G_N. \tag{9}$$

Notice that the scalar field does not couple to photons as $\delta T = 0$; hence, matter particles deviated with a larger Newtonian interaction than photons, $G_{\text{eff}} \geq G_N$. As a result, the modification of $G_N$ into $G_{\text{eff}}$ is not just a global rescaling and gravitational physics is genuinely modified. This appears, for instance, in the Shapiro effect (the time delay of photons in the presence of a massive object) as measured by the Cassini probe around the Sun. When the mass of the scalar field cannot be neglected, the effective Newton constant becomes distance-dependent:

$$G_{\text{eff}} = \left(1 + \frac{2\beta^2(\phi_0)}{Z(\phi_0)}e^{-m(\phi_0)r}\right)G_N, \tag{10}$$

where $r$ is the distance to the object sourcing the field. This equation allows us to classify the screening mechanisms.

### 2.3. The Non-Derivative Screening Mechanisms: Chameleon and Damour–Polyakov

The first mechanism corresponds to an environment-dependent mass $m(\phi_0)$. If the mass increases sharply inside dense matter, the scalar field emitted by any mass element deep inside a compact object is strongly Yukawa suppressed by the exponential term $e^{-m_\phi(\phi_0)r}$, where $r$ is the distance from the mass element. This implies that only a *thin shell* of mass $\Delta M$ at the surface of the object sources a scalar for surrounding objects to interact with. As a result, the coupling of the scalar field to this dense object becomes

$$\beta(\phi_0) \rightarrow \frac{\Delta M}{M}\beta(\phi_0) \tag{11}$$

where $M$ is the mass of the object. As long as $\Delta M/M \ll 1$, the effects of the scalar field are suppressed. This is the **chameleon mechanism** [48–51].

The second mechanism appears almost tautological. If in dense matter the coupling $\beta(\phi_0) = 0$, all small matter elements deep inside a dense object will not couple to the scalar field. As a result and similarly to the chameleon mechanism, only a thin shell over which the scalar profile varies at the surface of the objects interacts with other compact bodies. Hence the scalar force is also heavily suppressed. This is the **Damour–Polyakov mechanism** [52].

In fact, this classification can be systematised and rendered more quantitative using the effective field theory approach that we have advocated. Using Equation (7), we obtain

$$\frac{\varphi}{m_{\text{Pl}}} = \frac{2\beta(\phi_0)}{Z(\phi_0)}\Phi_N. \tag{12}$$

Let us first consider the case of a normalised scalar field with $Z(\phi_0) = 1$. The scalar field is screened when its response to the presence of an overdensity is suppressed compared to the Newtonian case. This requires that

$$\frac{|\varphi|}{2m_{\text{Pl}}|\Phi_N|} \leq \beta(\phi_{\text{out}}) \tag{13}$$

where $\varphi = \phi_{\text{in}} - \phi_0$ is the variation of the scalar field inside the dense object. Here $\Phi_N$ is the Newtonian potential at the surface of the object. This is the quantitative criterion for the chameleon and Damour–Polyakov mechanisms [48,53]. In particular, in objects that are sufficiently dense, the field $\phi_{\text{in}}$ nearly vanishes and $\varphi \simeq -\phi_0$ only depends on the environment. As a result, for such dense objects, screening occurs when $|\Phi_N| \geq \frac{\phi_0}{2m_{\text{Pl}}\beta(\phi_{\text{out}})}$, which depends only on the environment. Chameleon and Damour–Polyakov screenings occur for objects with a large enough surface Newtonian potential. In fact, it turns out that

$$\beta_A = \frac{|\varphi_A|}{2m_{\text{Pl}}|\Phi_N^A|} \tag{14}$$

for a screened object labelled by $A$ is the scalar charge of this object[3], i.e., its coupling to matter. The screening criterion (13) simply requires that the scalar charge of an object is less than the coupling of a test particle $\beta(\phi_0)$.

*2.4. The Derivative Screening Mechanisms: K-Mouflage and Vainshtein*

The third case, in fact, covers two mechanisms. If locally in a region of space, the normalisation factor

$$Z(\phi_0) \gg 1 \tag{15}$$

then obviously the effective coupling $\hat{\beta}(\phi_0) \ll \beta(\phi_0)$ and gravitational tests can be evaded. Notice that we define screening as reducing the effective coupling. This case covers the **K-mouflage**[4] and **Vainshtein** mechanisms.

The normalisation factor is a constant at the leading order. Going beyond leading order, i.e., including higher order operators in the effective field theory, $Z(\phi_0)$ can be expanded in a power series

$$Z(\phi_0) = 1 + a(\phi_0)r_c^2\frac{\Box\varphi}{m_{\text{Pl}}} + b(\phi_0)\frac{(\partial\varphi)^2}{\Lambda^4} + c(\phi)\frac{\Box^2\varphi}{\Lambda^5} + \dots \tag{16}$$

where $r_c$ is a cross over scale and has the dimension of length and $m_{\text{Pl}}$ is the Planck scale. The scale $\Lambda$ plays the role of the strong coupling scale of the models. The functions $a, b$ and $c$ are assumed to be smooth and of order unity.

2.4.1. K-Mouflage

The K-mouflage screening mechanism [46,54,55] is at play when $Z(\phi_0) \geq 1$ and the term in $(\partial\varphi)^2/\Lambda^4$ dominates in (16), i.e.,

$$|\vec{\nabla}\varphi| \geq \Lambda^2 \tag{17}$$

and therefore, the Newtonian potential must satisfy

$$|\vec{\nabla}\Phi_N| \geq \frac{\Lambda^2}{2\beta(\phi_0)m_{\text{Pl}}}. \tag{18}$$

Hence, K-mouflage screening occurs where the gravitational acceleration $\vec{a}_N = -\vec{\nabla}\Phi_N$ is large enough. Let us consider two typical situations. First, the Newtonian potential of a pointlike source of mass $M$ has a gradient satisfying (18) inside a radius $R_K$

$$R_K = \left(\frac{\beta(\phi_0)M}{4\pi m_{\text{Pl}}\Lambda^2}\right)^{1/2}. \tag{19}$$

The scalar field is screened inside the K-mouflage radius $R_K$. Another interesting example is given by the large scale structures of the Universe where the Newtonian potential is

sourced by overdensities $\delta\rho$ compared to the background energy density $\bar{\rho}$. In this case, screening takes place for wave-numbers $k$ such that

$$\beta(\phi_0)\delta \leq \frac{\Lambda^2 m_{\mathrm{Pl}} k}{\bar{\rho}} \tag{20}$$

where $\delta \equiv \frac{\delta\rho}{\bar{\rho}}$. In particular for models motivated by dark energy $\Lambda^4 \simeq m_{\mathrm{Pl}}^2 H_0^2$ screening occurs on scales such that $k/H_0 \gtrsim \beta(\phi_0)\delta$, i.e., large scale structures such as galaxy clusters are not screened as they satisfy $k/H_0 \lesssim 1$ [56,57].

2.4.2. Vainshtein

The Vainshtein mechanism [58,59] follows the same pattern as K-mouflage. The main difference is that now the dominant term in $Z(\phi_0)$, i.e., (16), is given by the $\Box\varphi$ term. This implies that

$$\Delta\Phi_N \geq \frac{1}{2\beta(\phi_0)r_c^2}, \tag{21}$$

i.e., screening occurs in regions where the spatial curvature is large enough. Taking once again a point source of mass $M$, the Vainshtein mechanism is at play on scales smaller than the Vainshtein radius.[5]

$$R_V = \left(\frac{3\beta(\phi_0)r_c^2}{4\pi m_{\mathrm{Pl}}^2 M}\right)^{1/3}. \tag{22}$$

Notice the power $1/3$ compared to the $1/2$ in the K-mouflage case. Similarly, on large scales where the density contrast is $\delta$, the scalar field is screened for wave numbers such that

$$\delta \geq \frac{1}{3\Omega_m\beta(\phi_0)} \tag{23}$$

where $\Omega_m$ is the matter fraction of the Universe when $r_c = 1/H_0$. The Vainshtein mechanism is stronger than K-mouflage and screens all structures reaching the non-linear regime $\delta \gtrsim 1$ as long as $\beta(\phi_0) \gtrsim 1$.

Finally, let us consider the case where the term in $\Box^2\varphi$ dominates in (16). This corresponds to[6]

$$\Delta^2\Phi_N \geq \frac{\Lambda^5}{2\beta(\phi_0)m_{\mathrm{Pl}}} \tag{24}$$

For a point source, the transition happens at the radius

$$R_{\mathrm{MG}} = \left(\frac{5M}{4\pi m_{\mathrm{Pl}}\Lambda^5}\right)^{1/5}. \tag{25}$$

As expected, the power is now $1/5$, which can be obtained by power counting. This case is particularly relevant as this corresponds to massive gravity and the original investigation by Vainshtein. In the massive gravity case [60,61]

$$\Lambda^5 = m_{\mathrm{Pl}} m_G^4 \tag{26}$$

where $m_G$ is the graviton mass.

In all these cases, screening occurs in a regime where one would expect the effective field theory to fail, i.e., when certain higher-order operators start dominating. Contrary to expectation, this is not always beyond the effective field theory regime. Indeed, scalar field theories with derivative interactions satisfy non-renormalisation theorems, which guarantee that these higher-order terms are not corrected by quantum effects [62,63]. Hence, the classical regime where some higher-order operators dominate can be trusted. This is in general not the case for non-derivative interaction potentials, which are corrected by

quantum effects. As a result, the K-mouflage and Vainshtein mechanisms appear more robust than the chameleon and Damour–Polyakov ones under radiative effects.

### 2.5. Screening Criteria: The Newtonian Potential and Its Derivatives

Finally, let us notice that the screening mechanisms can be classified by inequalities of the type

$$(\nabla^k)\Phi_N \gtrsim C \tag{27}$$

where $C$ is a dimensionful constant and $k = 0$ for chameleons, $k = 1$ for K-mouflage and $k = 2$ for the Vainshtein mechanism. This implies that it is the Newtonian potential, acceleration and space-time curvature, respectively, that govern objects' degrees of screening in these models. The case $k = 4$ appears for massive gravity. Of course, if higher-order terms in the expansion of $Z(\phi_0)$ in powers of derivatives were dominant, larger values of $k$ could also be relevant. As we have seen, from an effective field theory point of view, the powers $k = 0, 1, 2$ are the only ones to be considered. The case of massive gravity $k = 4$ only matters as the other cases $k \leq 2$ are forbidden due to the diffeomorphism invariance of the theory, see the discussion in Section 2.7.1.

### 2.6. Disformally Induced Charge

Let us now come back to a situation where the time dependence of the background is crucial. For future observational purposes, black holes are particularly important as the waves emitted during their collisions could carry much information about fundamental physics in previously untested regimes. For scalar fields mediating new interactions, this seems to be a perfect new playground. In most scalar field theories, no-hair theorems prevent the existence of a coupling between black holes and a scalar field, implying that black holes have no scalar charge (see Section 4.2 for observational consequences of this). However, these theorems are only valid in static configurations; in a time-dependent background, the black hole can be surrounded by a non-trivial scalar cloud.

Let us consider a canonically normalised and massless scalar field in a cosmological background. As before, we assume that locally Lorentz invariance is respected on the time scales under investigation. The Klein–Gordon equation becomes, in the presence of a static overdensity,

$$\Delta \varphi = \ddot{\gamma}(\phi_0)\rho. \tag{28}$$

As a result, we see that a scalar charge is induced by the cosmological evolution of the background [64,65]

$$\beta_{\text{ind}} = m_{\text{Pl}}(\gamma_2 \dot{\phi}_0^2 + \gamma_1 \ddot{\phi}_0) \tag{29}$$

where $\gamma_1 = \frac{d\gamma}{d\phi}$ and $\gamma_2 = \frac{d^2\gamma}{d\phi^2}$. This is particularly relevant to black holes solutions with a linear dependence in time $\dot{\phi}_0 = q$. In this case, the induced charge is strictly constant

$$\beta_{\text{ind}} = \gamma_2 m_{\text{Pl}} q^2 \tag{30}$$

which could lead to interesting phenomena in binary systems.

### 2.7. Examples of Screened Models

#### 2.7.1. Massive Gravity

The first description of screening in a gravitational context was given by Vainshtein and can be easily described using the Fierz–Pauli [60] modification of General Relativity (GR). In GR and in the presence of matter represented by the energy-momentum tensor $T^{\mu\nu}$, the response of the weak gravitational field $h_{\mu\nu} = g_{\mu\nu} - \eta_{\mu\nu}$ is given in momentum space by [7]

$$h_{\mu\nu}(p^\lambda) = \frac{16\pi G_N}{p^2}\left(T^{\mu\nu} - \eta_{\mu\nu}\frac{T}{2}\right) \tag{31}$$

where two features are important. The first is that $1/p^2 = 1/p^\lambda p_\lambda$ is characteristic of the propagation of a massless graviton. The second is the $1/2$ factor, which follows from the existence of two propagating modes. When the graviton becomes massive, the following mass term is added

$$\mathcal{L}_{\text{FP}} = -\frac{m_G^2 m_{\text{Pl}}^2}{8} (h_{\mu\nu} h^{\mu\nu} - h^2). \tag{32}$$

The tensorial structure of the Fierz–Pauli mass term guarantees the absence of ghosts in a flat background. The response to matter becomes

$$h_{\mu\nu}(p^\lambda) = \frac{16\pi G_N}{p^2 + m_G^2} \left( T^{\mu\nu} - \eta_{\mu\nu} \frac{T}{3} \right). \tag{33}$$

The factor in $1/(p^2 + m_G^2)$ is the propagator of a massive field of mass $m_G$. More surprising is the change $1/2 \rightarrow 1/3$ in the tensorial expression. In particular, in the limit $m_G \rightarrow 0$ one does not recover the massless case of GR. This is the famous vDVZ (van Dam-Veltman–Zakharov) discontinuity [66,67]. Its origin can be unravelled as follows. Writing

$$h_{\mu\nu} = \bar{h}_{\mu\nu} + \frac{\beta}{m_{\text{Pl}}} \varphi \eta_{\mu\nu} \tag{34}$$

where

$$\bar{h}_{\mu\nu}(p^\lambda) = \frac{16\pi G_N}{p^2} (T^{\mu\nu} - \eta_{\mu\nu} \frac{T}{2}) \tag{35}$$

and

$$\varphi = \frac{\beta}{m_{\text{Pl}}} \frac{T}{p^2 + m_g^2} \tag{36}$$

corresponding to a scalar satisfying

$$\Box\varphi - m_G^2 \varphi = -\frac{\beta}{m_{\text{Pl}}} T, \tag{37}$$

we find that (33) is satisfied provided that

$$\beta = \frac{1}{\sqrt{6}}. \tag{38}$$

Hence, we have decomposed the massive graviton into a helicity two part $\bar{h}_{\mu\nu}$ and a scalar part $\varphi$ coupled to matter with a scalar charge $\beta = 1/\sqrt{6}$. These are three of the five polarisations of a massive graviton. Notice that the scalar polarisation is always present however small the mass $m_G$, i.e., the massless limit is discontinuous as the number of propagating degrees of freedom is not continuous. As it stands, massive gravity with such a large coupling and a mass experimentally constrained to be $m_G \leq 10^{-22}$ eV would be excluded by solar system tests. This is not the case thanks to the Vainshtein mechanism.

Indeed non-linear interactions must be included as GR is not a linear theory. At the next order, one expects terms in $h^3$ leading to Lagrangian interactions of the type [61]

$$\mathcal{L}_3 \sim \frac{(\Box\varphi)^3}{\Lambda^5} \tag{39}$$

where $\Lambda^5 = m_{\text{Pl}} m_G^4$. The structure in $\Box\varphi$ follows from the symmetry $\varphi \rightarrow \varphi + \lambda_\mu x^\mu$, which can be absorbed into a diffeomorphism $\bar{h}_{\mu\nu} \rightarrow \bar{h}_{\mu\nu} + \partial_\mu \xi_\nu + \partial_\nu \xi_\mu$ where $\xi_\mu = \frac{\beta}{4m_{\text{Pl}}}(-2(\lambda.x)x_\mu + \lambda_\mu x^2)$. The Klein–Gordon equation is modified by terms in $\Box((\Box\varphi)^2)$. As a result, the normalisation factor is dominated by $Z(\phi_0) \sim \Box^2\varphi/\Lambda^5$ as mentioned in the previous section. This leads to the Vainshtein mechanism inside $R_{\text{MG}}$, which allows massive gravity to evade gravitational tests in the solar system, for instance.

### 2.7.2. Cubic Galileon Models

The cubic Galileon models [59] provide an example of a Vainshtein mechanism with the 1/5 power instead of the 1/3. They are defined by the Lagrangian

$$\mathcal{L}_{\text{Gal3}} = \frac{1}{2}(\partial\phi)^2 + \frac{(\partial\phi)^2 \Box\phi}{\Lambda^3}. \tag{40}$$

The normalisation factor for the kinetic terms involves $\Box\phi$ as expected. These theories are amongst the very few Horndeski models, which do not lead to gravitational waves with a speed differing from the speed of light. Unfortunately, as theories of self-accelerating dark energy, i.e., models where the acceleration is not due to a cosmological constant, they suffer from an anomalously large Integrated-Sachs-Wolfe (ISW) effect in the Cosmic Microwave Background (CMB). See Section 2.8 for more details.

### 2.7.3. Quartic K-Mouflage

The simplest example of K-mouflage model is provided by the Lagrangian [64]

$$\mathcal{L}_{\text{KM4}} = \frac{1}{2}(\partial\phi)^2 - \frac{(\partial\phi)^4}{\Lambda^4} \tag{41}$$

which is associated with a normalisation factor containing a term in $(\partial\phi)^2$. These models pass the standard tests of gravity in the solar system but need to be modified to account for the very small periastron anomaly of the Moon orbiting around the Earth. See Section 2.9 for more details.

### 2.7.4. Ratra–Peebles and f(R) Chameleons

Chameleons belong to a type of scalar-tensor theory [68] specified entirely by two functions of the field. The first one is the interaction potential $V(\phi)$ and the second one is the coupling function $A(\phi)$. The dynamics are driven by the effective potential [48,49]

$$V_{\text{eff}}(\phi) = V(\phi) + (A(\phi) - 1)\rho \tag{42}$$

where $\rho$ is the conserved matter density. When the effective potential has a minimum $\phi_0 \equiv \phi(\rho)$, its second derivative defines the mass of the chameleon

$$m^2(\rho) = \frac{d^2 V_{\text{eff}}}{d\phi^2}\big|_{\phi(\rho)} \tag{43}$$

Cosmologically, the chameleon minimum of the effective potential is an attractor when $m(\rho) \gg H$, i.e., the mass is greater than the Hubble rate [50]. This is usually guaranteed once the screening of the solar system has been taken into account, see Section 2.9. A typical example of chameleon theory is provided by [48,49]

$$V(\phi) = \frac{\Lambda^{n+4}}{\phi^n}, \quad A(\phi) = e^{\beta\phi/m_{\text{Pl}}} \tag{44}$$

associated with a constant coupling constant $\beta$. More generally, the coupling becomes density dependent as

$$\beta(\rho) = m_{\text{Pl}} \frac{d\ln A}{d\phi}\big|_{\phi(\rho)}. \tag{45}$$

Chameleons with $n = 1$ are extremely well constrained by laboratory experiments, see Section 3.5.

Surprisingly, models of modified gravity defined by the Lagrangian [34]

$$\mathcal{L}_{f(R)} = \frac{\sqrt{-g}f(R)}{16\pi G_N} \tag{46}$$

which is a function of the Ricci scalar $R$, can be transformed into a scalar-tensor setting. First of all, the field equations of $f(R)$ gravity can be obtained after a variation of the Lagrangian (46) with respect to the metric $g_{\mu\nu}$ and they read

$$f_R R_{\mu\nu} - \frac{1}{2} f g_{\mu\nu} - \nabla_\mu \nabla_\nu f_R + g_{\mu\nu} \Box f_R = 8\pi G T_{\mu\nu} \,, \tag{47}$$

where $f_R \equiv df(R)/dR$ and $\Box$ is the d'Alembertian operator. These equations naturally reduce to Einstein's field equations when $f(R) = R$. This theory can be mapped to a scalar field theory via

$$\frac{df}{dR} = e^{-2\beta\phi/m_{\rm Pl}} \tag{48}$$

where $\beta = \frac{1}{\sqrt{6}}$. The coupling function is given by the exponential

$$A(\phi) = e^{\beta\phi/m_{\rm Pl}} \tag{49}$$

leading to the same coupling to matter as massive gravity. Contrary to the massive gravity case, $f(R)$ models evade solar system tests of gravity thanks to the chameleon mechanism when the potential

$$V(\phi) = \frac{m_{\rm Pl}^2}{2} \frac{R \frac{df}{dR} - R}{(\frac{df}{dR})^2} \tag{50}$$

is appropriately chosen.

A popular model has been proposed by Hu–Sawicki [69] and reads

$$f(R) = -2\Lambda - \frac{f_{R_0} c^2}{n} \frac{R_0^{n+1}}{R^n} \,, \tag{51}$$

which involves two parameters, the exponent (positive definite) $n > 0$ and the normalisation $f_{R_0}$, which is constrained to be $|f_{R_0}| \lesssim 10^{-6}$ by the requirement that the solar system is screened [69] (see Section 2.9). On large scales, structures are screened for which

$$\Phi_N \gtrsim \chi \equiv \frac{3}{2} |f_{R_0}| \tag{52}$$

for the $n = 1$ Hu–Sawicki model, where $\chi$ is the "self-screening parameter". This follows directly from the fact that

$$f_{R_0} = -\frac{2\beta\delta\phi}{m_{\rm Pl}} \tag{53}$$

where $\delta\phi$ is the variation of the scalaron due to a structure in the present Universe. Assessing the inequality in (52)—or equivalently requiring that the scalar charge $Q = \frac{|\delta\phi|}{2\beta m_{\rm Pl} \Phi_N}$ must be less than $\beta = 1/\sqrt{6}$—gives a useful criterion for identifying unscreened objects (see Section 4).

### 2.7.5. $f(R)$ and Brans-Dicke

The $f(R)$ models can be written as a scalar-tensor theory of the Brans–Dicke type. The first step is to replace the $f(R)$ Lagrangian density by

$$\mathcal{L} = \sqrt{-g} [f(\lambda) + (R - \lambda) \frac{df(\lambda)}{d\lambda}] \tag{54}$$

which reduces to the original model by solving for $\lambda$. Then an auxiliary field

$$\psi \equiv \frac{df(\lambda)}{d\lambda} \tag{55}$$

can be introduced, together with the potential $V(\psi) = m_{\text{Pl}}^2(f(\lambda(\psi)) - \lambda(\psi)\psi)/2$, which corresponds to the Legendre transform of the function $f(\lambda)$. After replacing back into the original action, one recovers a scalar field action for $\psi$ in the Jordan frame that reads

$$S = \int \mathrm{d}^4x\sqrt{-g}\left[\psi R - \frac{\omega_{BD}(\psi)}{\psi}\nabla_\mu\psi\nabla^\mu\psi - V(\psi)\right] + 16\pi G\int \mathrm{d}^4x\sqrt{-\tilde{g}}\mathcal{L}_m(\psi_m^{(i)}, g_{\mu\nu}) .$$

(56)

This theory corresponds to the well-known generalised Jordan–Fierz–Brans–Dicke [70] theories with $\omega_{BD} = 0$. When the $\omega_{BD}$ parameter is non-vanishing and a constant, this reduces to the popular Jordan–Brans–Dicke theory. Exact solutions of these theories have been tested against observations of the Solar System [33,71], and the Cassini mission sets the constraint $\omega_{BD} > 40,000$, so that JBD has to be very close to GR. This bound is a reformulation of (88), see Section 2.9 for more details. After going to the Einstein frame, the theory must be a scalar-tensor with the Chameleon or Damour–Polyakov mechanisms in order to evade the gravitational tests in the solar system.

2.7.6. The Symmetron

The symmetron [72] is another scalar–tensor theory with a Higgs-like potential

$$V(\phi) = -\frac{\mu^2}{2}\phi^2 + \frac{\lambda}{4}\phi^4$$

(57)

and a non-linear coupling function

$$A(\phi) = 1 + \frac{\phi^2}{2M^2} + \dots.$$

(58)

where the quadratic term is meant to be small compared to unity. The coupling is given by

$$\beta(\phi) = \frac{m_{\text{Pl}}\phi}{M^2}$$

(59)

which vanishes at the minimum of the effective potential when $\rho > \mu^2 M^2$. This realises the Damour–Polyakov mechanism.

2.7.7. Beyond 4D: Dvali–Gabadadze–Porrati Gravity

The Dvali–Gabadadze–Porrati (DGP) gravity model [73] is a popular theory of modified gravity that postulates the existence of an extra fifth-dimensional Minkowski space, in which a brane of 3+1 dimensions is embedded. Its solutions are known to have two branches, one which is self-accelerating (sDGP), but is plagued with ghost instabilities [74] and another branch, the so-called normal branch (nDGP), which is non-self-accelerating and has better stability properties. At the non-linear level, the fifth-force is screened by the effect of the Vainshtein mechanism, and therefore, can still pass solar system constraints. This model can be written as a pure scalar-field model, and in the following, we will use the notations of [75] to describe the model and its cosmology. The action is given by

$$S = M_5^3\int d^5x\sqrt{-g}R + \int d^4x\sqrt{-g}\left[-2M_5^3K + \frac{M_4^2}{2}R - \sigma + \mathcal{L}_{\text{matter}}\right],$$

(60)

where $\mathcal{L}_{matter}$ is the matter Lagrangian, $R$ is the Ricci scalar built from the bulk metric $g_{ab}$ and $M_4$ and $M_5$ are the Planck scales on the brane and in the bulk, respectively. The metric $g_{\mu\nu}$ is on the brane, $R$ its Ricci scalar, and $K = g^{\mu\nu}K_{\mu\nu}$ is the trace of extrinsic curvature, $K_{\mu\nu}$. Finally, $\sigma$ is the tension or bare vacuum energy on the brane.

These two different mass scales give rise to a characteristic scale that can be written as

$$r_5 \approx \frac{M_4^2}{M_5^3}. \tag{61}$$

For scales larger than $r_5$, the five-dimensional physics contributes to the dynamics, while for scales smaller than $r_5$, gravity is four-dimensional and reduces to GR. The reader can find the complete set of field equations in [75]. After solving the Friedmann equations, the effective equation-of-state of this model is given by

$$w_{\text{eff}} = \frac{P}{\rho} = -\frac{\frac{dH}{dt} + 3H^2 + \frac{2\kappa}{a^2}}{3H^2 + \frac{3\kappa}{a^2}}, \tag{62}$$

where $\kappa$ is the three-dimensional spatial curvature. During the self-accelerating phase $w_{\text{eff}} \to -1$ in (62), therefore emulating a cosmological constant.

*2.8. Horndeski Theory and Beyond*

For four-dimensional scalar-tensor theories used so far, the action defining the system in the Einstein frame can be expressed as

$$S = \int \mathrm{d}^4x \sqrt{-g} \left[ \frac{m_{Pl}^2}{2} R - \frac{1}{2} (\nabla \phi)^2 - V(\phi) \right] + \int \mathrm{d}^4x \sqrt{-\tilde{g}} \mathcal{L}_m (\psi_m^{(i)}, \tilde{g}_{\mu\nu}) \tag{63}$$

where $\phi$ is the scalar field, $V(\phi)$, its potential and it couples to the matter fields $\psi_m^{(i)}$ through the Jordan frame metric $\tilde{g}_{\mu\nu}$, which is related to the metric $g_{\mu\nu}$ as

$$\tilde{g}_{\mu\nu} = A^2(\phi, X) g_{\mu\nu} + B^2(\phi, X) \partial_\mu \phi \partial_\mu \phi. \tag{64}$$

The disformal factor term in $B^2(\phi, X)$ leads to the derivative interactions in (2). In the previous discussions, see Section 2.7.4, we focused on the conformal parameter $A(\phi)$ chosen to be $X$-independent where $X = -(\partial \phi)^2/2\Lambda^2$ and $\Lambda$ is a given scale. Other choices are possible, which will dot be detailed here, in particular in the case of DHOST theories for which the dependence of $A(\phi, X)$ is crucial [45].

As can be expected, (63) can be generalised to account for all possible theories of a scalar field coupled to matter and the metric tensor. When only second-order equations of motion are considered, this theory is called the Horndeski theory. Its action can be written as

$$S = \int \mathrm{d}^4x \sqrt{-g} \left[ \sum_{i=2}^{5} \mathcal{L}_i + \mathcal{L}_m (\psi_m^{(i)}, g_{\mu\nu}) \right] \tag{65}$$

where the four Lagrangian terms correspond to different combinations of four functions $G_{2,3,4,5}$ of the scalar field and its kinetic energy $\chi = -\partial^\mu \partial_\mu \phi/2$, the Ricci scalar and the Einstein tensor $G_{\mu\nu}$ and are given by

$$
\begin{aligned}
\mathcal{L}_2 &= K(\phi, \chi), \\
\mathcal{L}_3 &= -G_3(\phi, \chi) \Box \phi, \\
\mathcal{L}_4 &= G_4(\phi, \chi) R + G_{4,\chi} \left[ (\Box \phi)^2 - (\nabla_\mu \nabla_\nu \phi)(\nabla^\mu \nabla^\nu \phi) \right], \\
\mathcal{L}_5 &= G_5(\phi, \chi) G_{\mu\nu} (\nabla^\mu \nabla^\nu \phi) \\
&\quad - \frac{1}{6} G_{5,\chi} \left[ (\Box \phi)^3 - 3(\Box \phi)(\nabla_\mu \nabla_\nu \phi)(\nabla^\mu \nabla^\nu \phi) \right. \\
&\quad + \left. 2(\nabla^\mu \nabla_\alpha \phi)(\nabla^\alpha \nabla_\beta \phi)(\nabla^\beta \nabla_\mu \phi) \right],
\end{aligned}
\tag{66}
$$

After the gravitational wave event GW170817 ([76,77], and as already anticipated in [78], the propagation of gravitational waves is practically equal to the speed of light, implying

that a large part of Horndeski theory with cosmological effects is ruled out, leaving mostly only models of type $\mathcal{L}_2$ and Cubic Galileons (Horndeski with Lagrangians up to $\mathcal{L}_3$) as the surviving class of models [79–81]. They are the ones that will be dealt with in this review and can be linked most directly to the screening mechanisms described here. When going beyond the Horndeski framework [82], the Vainshtein mechanism can break within massive sources [83]. This phenomenology was studied further in [84], and may be used to constrain such theories, as described in Section 4.1.

*2.9. Solar System Tests*

Screening mechanisms have been primarily designed with solar system tests in mind. Indeed light scalar fields coupled to matter should naturally violate the gravitational tests in the solar system as long as the range of the scalar interaction, i.e., the fifth force, is large enough and the coupling to matter is strong enough. The first and most prominent of these tests is provided by the Cassini probe [33] and constrains the effective coupling between matter and the scalar to be

$$\beta_{\mathrm{eff}}^2 \leq 4 \cdot 10^{-5} \tag{67}$$

as long as the range of the scalar force exceeds several astronomical units and $\beta_{\mathrm{eff}}^2$ corresponds to the strength of the fifth force acting on the satellite. As we have mentioned, this translates into the effective bound

$$\frac{\beta(\phi_0)\beta_\odot}{Z(\phi_0)} \leq 10^{-5} \tag{68}$$

where $\phi_0$ is the value of the scalar field in the interplanetary medium of the solar system. Here we have assumed that the Cassini satellite is not screened and the Sun is screened. As a result, the scalar charges are, respectively, the background one $\beta(\phi_0)$ for the satellite and $\beta_\odot$ for the Sun. In the case of the K-mouflage and Vainshtein mechanisms, the scalar charges of the Sun and the satellite are equal, and the Cassini bound can be achieved thanks to a large $Z(\phi_0)$ factor. As an example, for cubic Galileon models, the ratio between the fifth force and the Newtonian force behaves such as

$$\frac{F_N}{F_\phi} \simeq 2\beta^2 \left( \frac{r}{R_V} \right)^{3/2} \tag{69}$$

where $\beta(\phi_0) = \beta$ and $R_V$ is the Vainshtein radius. For cosmological models where $L \sim H_0^{-1}$, the Vainshtein radius of the Sun is around 0.1 kpc. As a result, for the planets of the solar system, $r/r_V \ll 1$ and the fifth force is negligible. K-mouflage models of cosmological interest with $\Lambda \simeq \Lambda_{\mathrm{DE}} \sim 10^{-3}$ eV lead to the same type of phenomenology with a K-mouflage radius of the Sun larger than 1000 a.u. and therefore no fifth force effects in the solar system. For chameleon-like models, the Cassini constraint becomes

$$\beta_\odot \lesssim 10^{-5} \tag{70}$$

where we have assumed that $Z(\phi_0) = 1$ and $\beta(\phi_0) \simeq 1$. This is a stringent bound, which translates into

$$\phi_0 \lesssim 10^{-11} m_{\mathrm{Pl}} \tag{71}$$

for the values of the scalar in the solar system. Indeed we have assumed that in dense bodies such as the Sun or planets, the scalar field vanishes. We have also used the Newtonian potential of the Sun $\Phi_{N\odot} \simeq 10^{-6}$.

In fact, chameleon-screened theories are constrained even more strongly by the Lunar Ranging experiment (LLR) [85,86]. This experiment constrains the Eötvos parameter

$$\eta_{AB} = \frac{|\vec{a}_A - \vec{a}_B|}{|\vec{a}_A + \vec{a}_B|} \tag{72}$$

for two bodies falling in the presence of a third one $C$. The accelerations $\vec{a}_{A,B}$ are towards $C$ and due to $C$. For bodies such as the Earth $A = \oplus$, the moon $B = $ moon and the Sun $C = \odot$, a non-vanishing value of the Eötvos parameter would correspond to a violation of the strong equivalence principle, i.e., a violation of the equivalence principle for bodies with a non-negligible gravitational self-energy. Such a violation is inherent to chameleon-screened models. Indeed, screened bodies have a scalar charge $\beta_A$, which is dependent on the Newtonian potential of the body $\beta_A \propto \Phi_A^{-1}$ implying a strong dependence on the nature of the objects. As the strength of the gravitational interaction between two screened bodies is given by

$$G_{AB} = G_N(1 + 2\beta_A\beta_B) \tag{73}$$

as long as the two objects are closer than the background Compton wavelength $m^{-1}(\phi_0)$, the Eötvos parameter becomes

$$\eta_{AB} \simeq \beta_C|\beta_A - \beta_B| \tag{74}$$

In the case of the LLR experiment, we have $\beta_A \simeq \phi_0/2\beta(\phi_0)\Phi_A m_{\text{Pl}}$ and therefore $\beta_{\text{moon}} \gg \beta_\oplus$. Using $\Phi_{N\text{moon}} \simeq 10^{-11}$ and $\Phi_\oplus \simeq 10^{-9}$ we find that the LLR constraint

$$\eta_{\text{LLR}} \leq 10^{-13}. \tag{75}$$

This becomes for the scalar charge of the Earth [49]

$$\beta_\oplus \lesssim 10^{-6} \tag{76}$$

which is stronger than the Cassini bound, i.e., we must impose that

$$\phi_0 \lesssim 10^{-15}m_{\text{Pl}}. \tag{77}$$

This corresponds to the energy scale of particle accelerators such as the Large Hadron Collider (LHC). This bound leads to a relevant constraint on the parameter space of popular models. Let us first consider the $n = 1$ inverse power-law chameleon model with

$$V(\phi) = \Lambda_{\text{DE}}^4 e^{\frac{\Lambda_{\text{DE}}}{\phi}} \approx \Lambda_{\text{DE}}^4 + \frac{\Lambda_{\text{DE}}^5}{\phi} + \dots. \tag{78}$$

This model combines the screening property of inverse power-law chameleon and the cosmological constant term $\Lambda_{\text{DE}}^4$ leading to the acceleration of the expansion of the Universe. The mass of the scalar is given by

$$m^2(\phi) \approx \frac{2\Lambda_{\text{DE}}^5}{\phi^3} \tag{79}$$

implying that in the solar system $m_0 \gtrsim 10^6 H_0$. Now, as long as the chameleon sits at the minimum of its effective potential, we have $m_{\text{cosmo}} \simeq (\frac{\rho_{\text{cosmo}}}{\rho_G})^{1/2} m_0$ where $\rho_{\text{cosmo}}$ is the cosmological matter density and $\rho_G \simeq 10^6 \rho_{\text{cosmo}}$ is the one in the Milky Way. As a result, we have the constraints on the cosmological mass of the chameleon [87,88]

$$m_{\text{cosmo}} \gtrsim 10^3 H_0. \tag{80}$$

As the Hubble rate is smaller than the cosmological mass, the minimum of the effective potential is a tracking solution for the cosmological evolution of the field. This bound (80) is generic for chameleon-screened models with an effect on the dynamics of the Universe on a large scale. In the context of the Hu–Sawicki model, and as $m_{\text{cosmo}}/H_0 \propto f_{R_0}^{-1/2}$, the solar system tests imply typically that $f_{R_0} \lesssim 10^{-6}$ [69]. For models with the Damour–Polyakov mechanism such as the symmetron, and if $\rho_G \leq \mu^2 M^2$, the field value in the solar system

is close to $\phi_0 \simeq \frac{\mu}{\sqrt{\lambda}}$. The mass of the scalar is also of order $\mu$, implying that the range of the symmetron is very short unless $\mu \lesssim 10^{-18}$ eV. In this case, the LLR bound applies and leads to

$$M \lesssim 10^{-3} m_{\text{Pl}} \tag{81}$$

which implies that the symmetron models must be an effective field theory below the grand unification scale.

Models with derivative screening mechanisms such as K-mouflage and Vainshtein do not violate the strong equivalence principle but lead to a variation of the periastron of objects such as the Moon [89]. Indeed, the interaction potential induced by a screening object does not vary as $1/r$ anymore. As a result, Bertrand's theorem[8] is violated and the planetary trajectories are not closed anymore. For K-mouflage models defined by a Lagrangian $\mathcal{L} = \Lambda^4 K(X)$ where $X = -(\partial\phi)^2/2\Lambda^4$ and $\Lambda \simeq \Lambda_{\text{DE}}$, the periastron is given by [90]

$$\delta\theta = 2\pi\beta^2 \frac{K'^2}{K''} x \frac{dX}{dx} \tag{82}$$

where $x = r/R_K$ is the reduced radius ($R_K$ is the K-mouflage radius). For the Moon, the LLR experiment implies that $\delta\theta \leq 2 \cdot 10^{-11}$ which constrains the function $K(X)$ and its derivatives $K'(X)$ and $K''(X)$. A typical example of models passing the solar system tests is given by $K(X) = -1 + X + K_\star(X - X_\star \arctan(\frac{X}{X_\star}))$ with $X_\star \simeq 1$ and $K_\star \gtrsim 10^3$. In these models, the screening effect is obtained as $K' \sim K_\star \gg 1$ as long as $|X| \gtrsim |X_\star|$. For cubic Galileons, the constraint from the periastron of the Moon reduces to a bound on the suppression scale [89,91]

$$\Lambda^3 \gtrsim m_{\text{Pl}} H_0^2. \tag{83}$$

The lower bound corresponds to Galileon models with an effect on cosmological scales.

Finally, models with the K-mouflage or the Vainshtein screening properties have another important characteristic. In the Jordan frame, where particles inside a body couple to gravity minimally, the Newton constant is affected by the conformal coupling function $A(\phi)$, i.e.

$$G_{\text{eff}} = A^2(\phi) G_N. \tag{84}$$

For chameleon-screened objects, the difference between the Jordan and Einstein values of the Newton constant is irrelevant as deep inside screened objects $\phi$ are constant and $A(\phi)$ can be normalised to be unity. This is what happens for symmetrons or inverse power-law chameleons, for instance. For models with derivative screening criteria, i.e., K-mouflage or Vainshtein, the local value of the field can be decomposed as

$$\phi(\vec{x}, t) \simeq \phi(\vec{x}) + \dot{\phi}_{\text{cosmo}}(t - t_0) + \phi_{\text{cosmo}}(t_0) \tag{85}$$

where $t_0$ is the present time. Here $\phi(\vec{x})$ is the value of the field due to the local and static distribution of matter whilst the correction term depends on time and follows from the contamination of the local values of the field by the large scale and cosmological variations of the field. In short, regions screened by the K-mouflage or Vainshtein mechanisms are not shielded from the cosmological time evolution of matter. As a result, the Newton constant in the Jordan frame becomes time-dependent with a drift [92]

$$\frac{d \ln G_{\text{eff}}}{dt} = \frac{\beta}{m_{\text{Pl}}} \dot{\phi} \tag{86}$$

where we have taken the scalar to be coupled conformally with a constant strength $\beta$. The LLR experiment has given a bound in the solar system [85]

$$\left| \frac{d \ln G_{\text{eff}}}{dt} \right| \leq 0.02 H_0, \tag{87}$$

i.e., Newton's constant must vary on timescales larger than the age of the Universe. This can be satisfied by K-mouflage or Vainshtein models provided $\beta \leq 0.1$ as long as $\dot{\phi} \sim m_{\mathrm{Pl}} H_0$, i.e., the scalar field varies by an order of magnitude around the Planck scale in one Hubble time [90].

### 3. Testing Screening in the Laboratory

Light scalar fields have a long range and could induce new physical effects in laboratory experiments. We will consider some typical experiments that constrain screened models in a complementary way to the astrophysical and cosmological observations discussed below. In what follows, the bounds on the screened models will mostly follow from the classical interaction between matter and the scalar field. A light scalar field on short enough scales could lead to quantum effects. As a rule, if the mass of the scalar in the laboratory environment is smaller than the inverse size of the experiment, the scalar can be considered to be massless. Quantum effects [93] of the Casimir type imply that two metallic plates separated by a distance $d$ will then interact and attract according to

$$F(d) = -\frac{\pi^2}{480 d^4} \tag{88}$$

as long as the coupling between the scalar and matter is large enough[9]. In the Casimir or Eötwash context, this would mean that the usual quantum effects due to electromagnetism would be increased by a factor of $3/2$. Such a large effect is excluded, and therefore, the scalar field cannot be considered as massless on the scales of these typical experiments. In the following, we will consider the case where the scalar is screened on the scales of the experiments, i.e., its typical mass is larger than the inverse of the size of the experimental setup. In this regime, where quantum effects can be neglected, the classical effects induced by the scalars are due to the non-trivial scalar profile and its non-vanishing gradient[10]. In the following, we will mostly focus on the classical case and the resulting constraints.

### 3.1. Casimir Interaction and Eötwash Experiment

We now turn to the Casimir effect [94], associated with the classical field between two metallic plates separated by a distance $d$. The classical pressure due to the scalar field with a non-trivial profile between the plates is attractive and with a magnitude given by [95]

$$\left| \frac{F_\phi}{A} \right| = V_{\mathrm{eff}}(\phi(0)) - V_{\mathrm{eff}}(\phi_0), \tag{89}$$

where $A$ is the surface area of the plates and $V_{\mathrm{eff}}$ is the effective potential. This is the difference between the potential energy in vacuum (i.e., without the plates) where the field takes the constant value $\phi_0$ and in the vacuum chamber halfway between the plates. In general, the field acquires a bubble-like profile between the plates and $\phi(0)$ is where the field is maximal. The density inside the plates is much larger than between the plates, so the field value inside the plates is zero to a very good approximation. For a massive scalar field of mass $m$ with a coupling strength $\beta$, the resulting pressure between two plates separated by distance $d$ is given by

$$\left| \frac{F_\phi}{A} \right| = \frac{\beta^2 \rho_{\mathrm{plate}}^2}{2 m_{\mathrm{Pl}}^2 m^2} e^{-md}, \tag{90}$$

which makes the Yukawa suppression of the interaction between the two plates explicit. In the screened case, the situation can be very different.

Let us first focus on the symmetron case. As long as $\mu \gtrsim d^{-1}$, the value $\phi(0)$ is very close to the vacuum value $\phi(0) \simeq \frac{\mu}{\sqrt{\lambda}}$ implying that $F_\phi/A \simeq 0$, i.e., the Casimir effect does not efficiently probe symmetrons with large masses compared to the inverse distance between the plates. On the other hand, when $\mu \lesssim d^{-1}$, the field essentially vanishes

in the plates and between the plates [96]. As a result, the classical pressure due to the scalar becomes

$$\left| \frac{F_\phi}{A} \right| = \frac{\mu^4}{4\lambda}. \tag{91}$$

Notice that in this regime, the symmetron decouples from matter inside the experiment as $\beta(\phi(0)) = 0$. We will see how this compares to the quantum effects in Section 3.6. We can now turn to the chameleon case where we assume that the density between the plates vanishes and is infinite in the plates. This simplified the expression of the pressure, which becomes [97]

$$\left| \frac{F_\phi}{A} \right| = \Lambda^4 \left( \frac{\sqrt{2} B(\frac{1}{2}, \frac{2+n}{2n})}{n \Lambda d} \right)^{2n/(2+n)}. \tag{92}$$

where $B(.\,,.)$ is Euler's $B$ function. In the chameleon case, the pressure is a power-law depending on $2n/(n+2)$, which can be very flat in $d$ and, therefore, dominates the photon Casimir pressure at large distances. Quantum effects can also be taken into account when the chameleon's mass is small enough, see Section 3.6.

The most stringent experimental constraint on the intrinsic value of the Casimir pressure has been obtained with a distance $d = 746$ nm between two parallel plates and reads $|\frac{\Delta F_\phi}{A}| \leq 0.35$ mPa [98]. The plate density is of the order of $\rho_{\text{plate}} = 10$ g cm$^{-3}$. The constraints deduced from the Casimir experiment can be seen in Section 3.5. It should be noted that realistic experiments sometimes employ a plate-and-sphere configuration, which can have an $O(1)$ modification to (92) [99].

The Eöt-Wash experiment [100] is similar to a Casimir experiment and involves two rotating plates separated by a distance $d$. Each plate is drilled with holes of radii $r_h$ spaced regularly on a circle. The gravitational and scalar interactions vary in time as the two plates rotate, hence inducing a torque between the plates. This effect can be understood by evaluating the potential energy of the configuration. The potential energy is obtained by calculating the amount of work required to approach one plate from infinity [35,101]. Defining by $A(\theta)$ the surface area of the two plates that face each other at any given time, a good approximation to energy is simply the work of the force between the plates corresponding to the amount of surface area in common between the two plates. The torque is then obtained as the derivative of the potential energy of the configuration with respect to the rotation angle $\theta$ and is given by

$$T \sim a_\theta \int_d^\infty dx \, \frac{F_\phi}{A}, \tag{93}$$

where $a_\theta = \frac{dA}{d\theta}$ depends on the experiment and is a well-known quantity. As can be seen, the torque is a direct consequence of the classical pressure between two plates.

For a Yukawa interaction and upon using the previous expression (89) for the classical pressure, we find that the torque is given by

$$T = a_\theta \, \frac{\beta^2 \rho_{\text{plate}}^2}{2 m_{\text{Pl}}^2 m^3} e^{-md}, \tag{94}$$

which is exponentially suppressed with the separation between the two plates $d$. Let us now consider the symmetron and chameleon cases. In the symmetron case, the classical pressure is non-vanishing only when $d \lesssim \mu^{-1}$, implying that

$$d \lesssim \mu^{-1}, \, T \simeq a_\theta \frac{\mu^4 d}{4\lambda}$$

$$d \gtrsim \mu^{-1}, T \simeq a_\theta \frac{\mu^3}{4\lambda}. \tag{95}$$

Hence, the torque increases linearly before saturating at a maximal value. For chameleons, three cases must be distinguished. First, when $n > 2$, the torque is insensitive to the long range behaviour of the chameleon field in the absence of the plates and we have

$$T = a_\theta \left( \frac{2+n}{n-2} \right) \Lambda^4 d \left( \frac{\sqrt{2} B \left( \frac{1}{2}, \frac{2+n}{2n} \right)}{n\Lambda d} \right)^{2n/(2+n)} \tag{96}$$

which decreases with the distance. In the case $n < 2$, the torque is sensitive to the Yukawa suppression of the scalar field at distances larger that $d_\star \sim m_0^{-1}$, where $m_0$ is the mass in the vacuum between the plates. This becomes

$$T \sim a_\theta \left( \frac{2+n}{n-2} \right) \Lambda^4 \left[ d_\star \left( \frac{\sqrt{2} B \left( \frac{1}{2}, \frac{2+n}{2n} \right)}{n\Lambda d_\star} \right)^{2n/(2+n)} - d \left( \frac{\sqrt{2} B \left( \frac{1}{2}, \frac{2+n}{2n} \right)}{n\Lambda d} \right)^{2n/(2+n)} \right] \tag{97}$$

for $d \lesssim d_\star$ and essentially vanishes for larger distances. In the case $n = 2$, a logarithmic behaviour appears.

The 2006 Eöt-Wash experiment [102] provided the bound for a separation between the plates of $d = 55$ μm, which is

$$|T| \leq a_\theta \Lambda_T^3, \tag{98}$$

where $\Lambda_T = 0.35\Lambda_{DE}$ [35] and $\Lambda_{DE} = 2.4$ meV. We must also modify the torque calculated previously in order to take into account the effects of a thin electrostatic shielding sheet of width $d_s = 10$ μm between the plates in the Eöt-Wash experiment. This reduces the observed torque, which becomes $T_{obs} = e^{-m_c d_s} T$. As a result, we have that

$$e^{-m_c d_s} \int_{55\mu m}^{\infty} \left| \frac{F_\phi}{A} \right| dx \leq \Lambda_T^3. \tag{99}$$

Surprisingly, the Eötwash experiment tests the dark energy scale in the laboratory as $\Lambda_T \approx \Lambda_{DE}$.

*3.2. Quantum Bouncer*

Neutrons behave quantum mechanically in the terrestrial gravitational field. The quantised energy levels of the neutrons have been observed in Rabi oscillation experiments [103]. Typically a neutron is prepared in its ground state by selecting the width of its wave function using a cache, then a perturbation induced either mechanically or magnetically makes the neutron state jump from the ground state to one of its excited levels. Then the ground state is again selected by another cache. The missing neutrons are then compared with the probabilities of oscillations from the ground state to an excited level. This allows one to detect the first few excited states and measure their energy levels. Now, if a new force complements gravity, the energy levels will be perturbed. Such perturbations have been investigated and typically the bounds are now at the $10^{-14}$ eV level.

The wave function of the neutron satisfies the Schrödinger equation

$$\left( -\frac{\hbar^2}{2m_N} \frac{d^2}{dz^2} + V(z) \right) \psi_n = E_n \psi_n \tag{100}$$

where $m_N$ is the neutron's mass and the potential over a horizontal plate is

$$V(z) = m_N g z + m_N (A(\phi(z)) - 1) \tag{101}$$

where $\phi(z)$ is the vertical profile of the scalar field. We put the mirror at $z \leq 0$. The contribution due to the scalar field is

$$\delta V(z) = m_N (A(\phi(z)) - 1) \tag{102}$$

which depends on the model. In the absence of any scalar field, the wavefunctions are Airy functions

$$\psi_k(z) = c_k \text{Ai}\left(\frac{z}{z_0} - \epsilon_k\right) \tag{103}$$

where $c_k$ is a normalisation constant, $z_0 = (\hbar^2/2m_N^2 g)^{1/3}$, $-\epsilon_k$ are the zeros of the Airy function. Typically $\epsilon_k = \{2.338, 4.088, 5.521, 6.787, 7.944, 9.023 \dots\}$ for the first levels $k = 1\dots$. At the first-order of perturbation theory, the energy levels are

$$E_k = E_k^{(0)} + \delta E_k \tag{104}$$

where $E_k^{(0)} = \epsilon_k m_N g z_0$, and the perturbed energy level

$$\delta E_k = m_N \langle \psi_k | (A(\phi(z)) - 1) | \psi_k \rangle \tag{105}$$

is the averaged value of the perturbed potential in the excited states.

Let us see what this entails for chameleon models [104,105]. In this case, the perturbation depends on

$$A(\phi) - 1 \simeq \frac{\beta}{m_{\text{Pl}}}\phi + \dots \tag{106}$$

where the profile of the chameleon over the plate is given by

$$\phi(z) = \Lambda\left(\frac{2+n}{\sqrt{2}}\Lambda z\right)^{2/(n+2)}. \tag{107}$$

Using this form of the correction to the potential energy, i.e., power laws, and the fact that the corrections to the energy levels are linear in $\beta$, one can deduce useful constraints on the parameters of the model. Thus far, we have assumed that the neutrons are not screened. When they are screened, the correction to the energy levels is easily obtained by replacing $\beta \to \lambda\beta$ where $\lambda$ is the corresponding screening factor.

In the case of symmetrons, the correction to the potential energy depends on

$$A(\phi) - 1 = \frac{\phi^2}{2M^2} \tag{108}$$

whilst the symmetron profile is given by [106]

$$\phi(z) = \frac{\mu}{\sqrt{\lambda}} \tanh \frac{\mu z}{\sqrt{2}} \tag{109}$$

where we assume that the plate is completely screened. The averaged values of $\delta V$ are constrained by

$$|\delta E_3 - \delta E_1| \lesssim 2 \cdot 10^{-15} \text{ eV} \tag{110}$$

which leads to strong constraints on symmetron models. See Section 3.5.

### 3.3. Atomic Interferometry

Atomic interferometry experiments are capable of very precisely measuring the acceleration of an atom in free fall [107,108]. By placing a source mass in the vicinity of the atom and performing several measurements with the source mass in different positions, the force $F_{\text{source}}$ between the atom and the source mass can be isolated. That force is a sum of both the Newtonian gravitational force and any heretofore undiscovered interactions:

$$\vec{F}_{\text{source}} = \vec{F}_{\text{N}} + \vec{F}_\phi . \tag{111}$$

As such, atom interferometry is a sensitive probe of new theories that predict a classical fifth force $\vec{F}_\phi$. In experiments such as [109,110] the source is a ball of matter and the

extra acceleration $a_\phi = F_\phi / m_{\text{atom}}$ is determined at the level $a_\phi \lesssim 5.5\ \mu\text{m}/\text{s}^2$ at a distance $d = R_B + d_B$ where $R_B = 0.95$ cm is the radius of the ball and $d_B = 0.88$ cm is the distance to the interferometer. The whole setup is embedded inside a cavity of radius $R_c = 6.1$ cm.

Scalar fields, of the type considered in this review, generically predict such a force. The fifth force is of the form

$$\vec{F}_\phi = -m_{\text{atom}} \vec{\nabla} \ln A(\phi) \,, \tag{112}$$

where $A(\phi)$ is the coupling function to matter. In essence, the source mass induces a nonzero field gradient producing a fifth force, allowing atom interferometry to test scalar field theories.

The fifth force depends on the scalar charge $q_A$ of the considered object $A$, i.e., on the way an object interacts with the scalar field. In screened theories, it is often written as the product $q_A = \beta_A m_A$ of the mass $m_A$ of the objects and the reduced scalar charge $\beta_A$. The reduced scalar charge can be factorised as $\beta_A = \lambda_A \beta(\phi_0)$ where $\beta(\phi_0)$ is a the coupling of a point-particle to the scalar field in the background environment characterised by the scalar field value $\phi_0$. The screening factor $\lambda_A$ takes a numerical value between 0 and 1 and in general depends on the strength and form of the scalar-matter coupling function $A(\phi)$, the size, mass, and geometry of the object, as well as the ambient scalar field value $\phi_0$. For a spherical object, the screening factor of object $A$ is given by

$$\lambda_A = \frac{|\phi_A - \phi_0|}{2 m_{\text{Pl}} \Phi_A} \tag{113}$$

when the object is screened, otherwise $\lambda_A = \beta(\phi_0)$. Here $\phi_A$ is the value of the scalar field deep inside the body $A$, $\Phi_A$ the Newtonian potential at its surface and $\phi_0$ is the ambient field value far away from the object. In terms of the screening factors, the force between two bodies $A, B$ is

$$|F_\phi| = \left( \frac{\beta(\phi_0)}{m_{\text{Pl}}} \right)^2 \frac{(\lambda_A m_a)(\lambda_B m_B)}{4\pi r^2} e^{-m_c r} \,, \tag{114}$$

where $m_c$ is the effective mass of the scalar particle's fluctuations. In screened theories, the screening factors of macroscopic objects are typically tiny, necessitating new ways to test gravity in order to probe the screened regime of these theories. Atom interferometry fits the bill perfectly [111,112], as small objects such as atomic nuclei are typically unscreened. Consequently, screened theories predict large deviations from Newtonian gravity inside those experiments. Furthermore, the experiment is performed in a chamber where the mass $m_0 = m(\phi_0)$ of the scalar particles is small, and distance scales of order $\sim$ cm are probed. The strongest bounds are achieved when the source mass is small, approximately the size of a marble, and placed inside the vacuum chamber, as a metal vacuum chamber wall between the bodies would screen the interaction.

Within the approximations that led to Equation (114), one only needs to determine the ambient field value $\phi_0$ inside the vacuum chamber. This quantity depends on the precise theory in question, but some general observations may be made. First, in a region with uniform density $\rho$, the field will roll to minimise its effective potential $V_{\text{eff}}(\phi)$ given by (42) for a value $\phi(\rho)$. In a dense region such as the vacuum chamber walls, $\phi(\rho)$ is small, while in the rarefied region inside the vacuum chamber $\phi(\rho)$ is large. The field thus starts at a small value $\phi_{\text{min,wall}}$ near the walls and rolls towards a large value $\phi_{\text{min,vac}}$ near the centre. However, the field will only reach $\phi_{\text{min,vac}}$ if the vacuum chamber is sufficiently large. The energy of the scalar field depends upon both potential energy $V(\phi)$ and gradient energy $(\vec{\nabla}\phi)^2$. A field configuration that rolls quickly to the minimum has relatively little potential energy but a great deal of gradient energy and vice-versa. The ground state classical field configuration is the one that minimises the energy and, hence, is a balance between the potential and field gradients. If the vacuum chamber is small, then the minimum energy

configuration balances these two quantities by rolling to a value such that the mass of the scalar field is proportional to the size $R$ of the vacuum chamber [49,53]

$$m_0(\phi_{\text{vac}}) \equiv \frac{1}{R} \, . \tag{115}$$

If the vacuum chamber is large, though, then there is plenty of room for the field to roll to the minimum of the effective potential. The condition for this to occur is

$$m_0(\phi_{\text{min,vac}}) \gg \frac{1}{R} \, . \tag{116}$$

As such, the field inside the vacuum chamber is

$$\phi_0 = \min(\phi_{\text{min,vac}}, \phi_{\text{vac}}) \, . \tag{117}$$

It should be noted that in practical experiments, where there can be significant deviations from the approximations used here, i.e., non-spherical source masses and an irregularly shaped vacuum chambers, numerical techniques have been used to solve the scalar field's equation of motion in three dimensions. This enables the experiments to take advantage of source masses that boost the sensitivity of the experiment to fifth forces by some 20% [110]. More exotic shapes have been shown to boost the sensitivity even further, by up to a factor $\sim 3$ [113].

Atom interferometry experiments of this type, with an in-vacuum source mass, have now been performed by two separate groups [109,114,115]. In these experiments, the acceleration between an atom and a marble-sized source mass has been constrained to $a \lesssim 50 \, \text{nm/s}^2$ at a distance of $r \lesssim 1$ cm. These experiments have placed strong bounds on the parameters of chameleon and symmetron [116] modified gravity, as will be detailed in Section 3.5.

*3.4. Atomic Spectroscopy*

In the previous section, we saw that the scalar field mediates a new force, Equation (114), between extended spherical objects. This same force law acts between atomic nuclei and their electrons, resulting in a shift of the atomic energy levels. Consequently, precision atomic spectroscopy is capable of testing the modified gravity models under consideration in this review.

The simplest system to consider is hydrogen, consisting of a single electron orbiting a single proton. The force law of Equation (114) perturbs the electron's Hamiltonian [117,118]

$$\delta H = \left( \frac{\beta(\phi_0)}{m_{\text{Pl}}} \right)^2 \frac{\lambda_p m_p m_e}{r} \, , \tag{118}$$

where $\lambda_p, m_p$ are the screening factor and mass of the proton, and we have assumed that the scalar field's Compton wavelength $m_c$ is much larger than the size of the atom. The electron is pointlike and is, therefore, unscreened.[11] The perturbation to the electron's energy levels are computed via the first-order perturbation theory result

$$\delta E_n = \langle \psi_n | \widehat{\delta H} | \psi_n \rangle \, , \tag{119}$$

where $\psi_n$ are the unperturbed electron's eigenstates.

This was first computed for a generic scalar field coupled to matter with a strength $\beta(\phi) = m_{\text{Pl}}/M$ [119], using measurements of the hydrogen 1s-2s transition [120–122] to rule out

$$M \lesssim \text{TeV} \, . \tag{120}$$

However, that study did not account for the screening behaviour exhibited by chameleon and symmetron theories. That analysis was recently extended to include screened theories and the work will be published shortly, resulting in the bound that is illustrated in Figure 1.

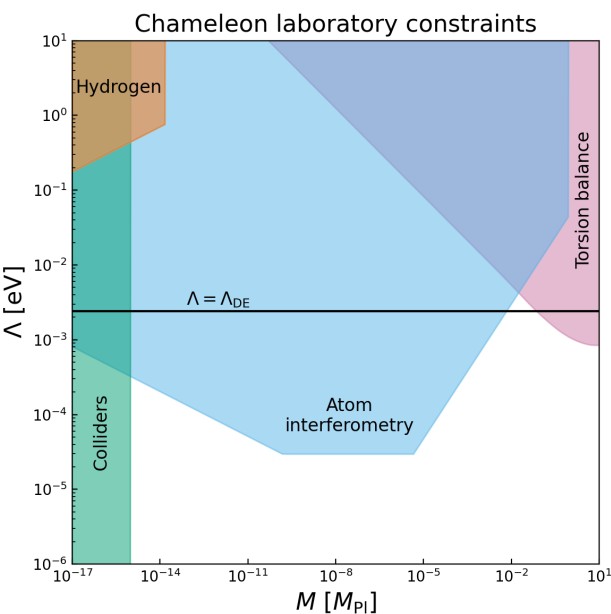

**Figure 1.** Current bounds on $n = 1$ chameleon theory parameters from various experiments.

### 3.5. Combined Laboratory Constraints

Combined bounds on theory parameters derived from the experimental techniques detailed in this section are plotted in Figures 1 and 2. Chameleons and symmetrons have similar phenomenology and, hence, are constrained by similar experiments. Theories exhibiting Vainshtein screening, however, are more difficult to constrain with local tests, as the presence of the Earth or Sun nearby suppresses the fifth force. Such effects were considered in [91] and only restricted to planar configurations where the effects of the Earth are minimised.

The chameleon has a linear coupling to matter, often expressed in terms of a parameter $M = m_{\rm Pl}/\beta$. Smaller $M$ corresponds to a stronger coupling. Experimental bounds on the theory are dominated by three tests. At sufficiently small $M$, the coupling to matter is so strong that collider bounds rule out a wide region of parameter space. At large $M \gtrsim m_{\rm Pl}$, the coupling is sufficiently weak that even macroscopic objects are unscreened, so torsion balances are capable of testing the theory. In the intermediate range, the strongest constraints come from atom interferometry. One could also consider chameleon models with $n \neq 1$. In general, larger values of $n$ result in more efficient screening effects; hence, the plots on constraints would look similar but with weaker bounds overall.

The bounds on symmetron parameter space are plotted in Figure 2. Unlike the chameleon, the symmetron has a mass parameter $\mu$ that fixes it to a specific length scale $\mu^{-1}$. For an experiment at a length scale $L$, if $L \gg \mu^{-1}$ then the fifth force would be exponentially suppressed, as is clear in Equation (114). Likewise, in an enclosed experiment, if $L \ll \mu^{-1}$ then the energy considerations in the previous subsection imply that the field simply remains in the symmetric phase where $\phi = 0$. The coupling to matter is quadratic,

$$\mathcal{L}_{\rm symm} \supset -\frac{\beta(\phi)}{m_{\rm Pl}}\rho \equiv -\frac{\phi^2}{M^2}\rho \,, \tag{121}$$

so in the symmetric phase, where $\phi = 0$, the coupling to matter switches off and the fifth force vanishes. Therefore, to test a symmetron with mass parameter $\mu$, one must test it with an experiment on a length scale $L \approx \mu^{-1}$.

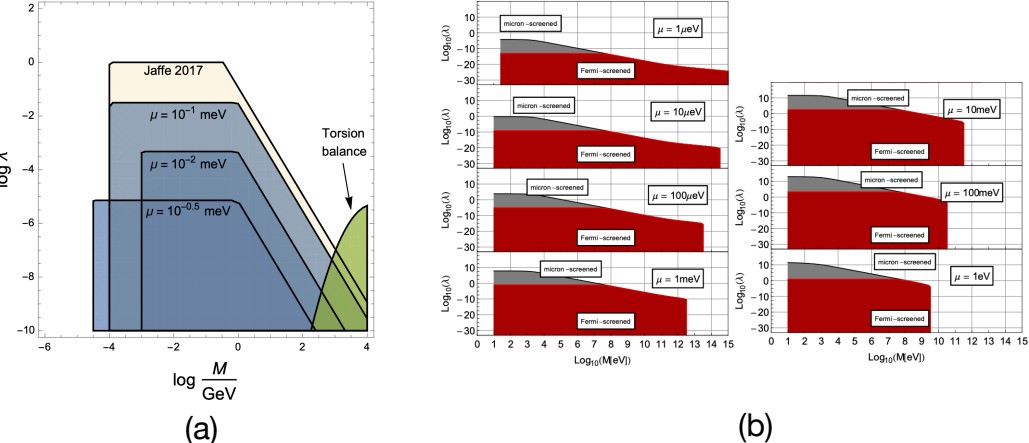

(a)                                                    (b)

**Figure 2.** Current bounds on symmetron theory parameters from atom interferometry and torsion balances (**a**) and from cold bouncing neutrons (**b**). The tan curves derive from atom interferometry, while the green region is ruled out by torsion balances. Both the tan and green regions apply only to $\mu = 10^{-1}$ meV. On the right, only the red curves have been conclusively ruled out by bouncing neutrons. Reproduced from [115,123].

### 3.6. Quantum Constraints

Classical physics effects induced by the light scalar field have been detailed so far. It turns out that laboratory experiments can also be sensitive to the quantum properties of the scalar field. This can typically be seen in two types of situations. In particle physics, the scalars are so light compared to accelerator scales that light scalars can be produced and have a phenomenology very similar to dark matter, i.e., they would appear as missing mass. They could also play a role in the precision tests of the standard model. As we already mentioned above, when the scalars are light compared to the inverse size of the laboratory scales, we can expect that they will induce quantum interactions due to their vacuum fluctuations. This typically occurs in Casimir experiments where two plates attract each other or the Eötwash setting where two plates face each other.

Particle physics experiments test the nature of the interactions of new states to the standard model at very high energy. In particular, the interactions of the light scalars to matter and the gauge bosons of the standard are via the Higgs portal, i.e., the Higgs field couples both to the standard model particles and the light scalar and as such mediates the interactions of the light scalar to the standard model. This mechanism is tightly constrained by the precision tests of the standard model. For instance, the light scalars will have an effect on the fine structure constant, the mass of the $Z$ boson or the Fermi interaction constant $G_F$. The resulting bound on $\beta = \frac{m_{\rm Pl}}{M}$ is [124]

$$M \gtrsim 10^3 \text{ GeV} \tag{122}$$

which tells us that the light scalar must originate from a completion at energies much larger than the standard model scale.

Quantum effects are also important when the light scalars are strongly coupled to the walls of the Casimir or the Eötwash experiment and light enough in the vacuum between the plates. The mass of the scalar field is given by

$$m^2 = V''(\phi) + A''(\phi)\rho. \tag{123}$$

The density is piece-wise constant and labelled $\rho_{1,2,3}$ in the case of a Casimir experiment. Here $\rho_2$ is the density between the plates. Notice that as $\phi$ is continuous, the mass jumps across boundaries as $\rho$ varies from the vacuum density to the plate one. The force between two objects can be calculated using a path integral formalism, which takes into account

both the classical effects already investigated in this review and the quantum effects akin to the Casimir interaction [125]

$$\vec{F}_\phi = -\int d^3x \, \vec{\partial}_d \rho \langle A(\phi) \rangle \tag{124}$$

where the integration is taken over all space and $\vec{\partial}_d \rho$ is the derivative in the direction defined by the parameter $d$, which specifies the position of one of the bodies. Varying $d$ is equivalent to changing the distance between the objects. For instance, in the case of a plate of density $\rho_3$ positioned along the $x$-axis between $x = d$ and $x = d + L$, the vacuum of density $\rho_2$ between $x = 0$ and $L$, a plate of density $\rho_1$ for $x \leq 0$ and finally again the vacuum for $x > d + L$ we have $\partial_d \rho = (\rho_3 - \rho_2)(\delta(x - d - L) - \delta(x - d))$ and the force is along the $x$-axis. The quantum average $\langle A(\phi) \rangle$ is taken over all the quantum fluctuations of $\phi$. When the field has a classical profile $\phi_{\text{clas}}$, this quantum calculation can be performed in perturbation theory

$$\langle A(\phi) \rangle = A(\phi_{\text{clas}}) + \frac{A''(\phi_{\text{clas}})}{2} \Delta(x, x) + \dots \tag{125}$$

The first contribution leads to the classical force that we have already considered. The second term is the leading quantum contribution. Notice that the linear coupling in $A'$ is absent as the quantum fluctuations involve the fluctuations around a background, which satisfies the equations of motion of the system. The higher-order terms in the expansion of $A(\phi)$ in a power series are associated with higher loop contributions to the force when the first term is given by a one-loop diagram. The Feynman propagator $\Delta(x, x)$ at coinciding points is fraught with divergences. Fortunately, they cancel in the force calculation as we will see.

Let us focus on the one-dimensional force as befitting Casimir experiments. The quantum pressure on a plate of surface area $A$ is then given by

$$\frac{F_x}{A} = -\frac{A''}{2}(\rho_3 - \rho_2)(\Delta(d, d) - \Delta(d + L, d + L)) \tag{126}$$

where we have considered that the derivative $A''(\phi_{\text{clas}}) \simeq A''$ is nearly constant. This is exact for symmetron models and chameleon models with $\phi \ll M$. As the classical solution is continuous at the boundary between the plates, the quantum force is in fact given by

$$\frac{F_x}{A} = \frac{m_2^2 - m_3^2}{2}(\Delta(d, d) - \Delta(d + L, d + L)) \tag{127}$$

where $m_3$ is the mass of the scalar close to the boundary and inside the plate, whereas $m_2$ is the mass close to the boundary and in the vacuum. As the quantum divergence of $\Delta(x, x)$ are $x$-independent, we see immediately that they cancel in the force (127), which is finite. Moreover, the limit $L \to \infty$ is finite and corresponds to the case of an infinitely wide plate. Notice that the contribution in $-\Delta(d + L, d + L)$ is the usual renormalisation due to the quantum pressure exerted to the right of the very wide plate of width $L$.

In the case of a Casimir experiment between two plates, the Feynman propagator with three regions (plate-vacuum-plate) must be calculated. In the case of the Eötwash experiment, where a thin electrostatic shield lies between the plate, the Feynman propagator is obtained by calculating a Green's function involving five regions. In practice, this can only be calculated analytically by assuming that the mass of the scalar field is nearly constant in each of the regions. This leads to the expression

$$\frac{F_x}{A} = -\frac{1}{2\pi^2} \int_0^\infty d\rho \rho^2 \frac{\gamma_2(\gamma_2 - \gamma_1)(\gamma_2 - \gamma_3)}{e^{2d\gamma_2}(\gamma_1 + \gamma_2)(\gamma_2 + \gamma_3) - (\gamma_2 - \gamma_1)(\gamma_2 - \gamma_3)} \tag{128}$$

with $\gamma_i^2 = \rho^2 + m_i^2$. When the density in the plates becomes extremely large compared to the one in the vacuum, the limit $m_{1,3} \to \infty$ gives the finite result

$$\frac{F_x}{A} = -\frac{1}{2\pi^2} \int_0^\infty d\rho \rho^2 \frac{\gamma_2}{e^{2d\gamma_2} - 1}.$$ (129)

For massless fields in the vacuum $m_2 = 0$, this gives the Casimir interaction (88) as expected.

When applying these results to screened models, care must be exerted as they assume that the mass of the field is constant between the plates. The quantum contributions to the pressure $F_x/A$ can be constrained by the Casimir experiments and the resulting torque between plates by the Eötwash results. These are summarised in Figure 3 for symmetrons. In a nutshell, when the $\mu$ parameter of the symmetron model becomes lower than $1/d$, the field typically vanishes everywhere. The linear coupling to matter vanishes but $A'' = 1/M^2$ is non-vanishing thus providing the quadratic coupling to the quantum fluctuations. As the density between the plate is small but nonzero, the mass of the scalar remains positive and the quantum calculation is not plagued with quantum instabilities. For chameleons, the coupling can be taken as $A'' \simeq 1/M^2$ too. The main difference is that when the density between the plates is low, the mass of the scalar cannot become much lower than $1/d$, see (116), implying that the quantum constraints are less strong than in the symmetron case.

As the expansion of $A(\phi)$ involves higher-order terms suppressed by the strong coupling scale $M$ and contributing to higher loops, they can be neglected on distances between the plates $d \gtrsim 1/\sqrt{m_{1,3}M}$. As the density in the plates is very large, this is always a shorter distance scale than $1/M$ where the calculations of the effective field theory should not be trusted naively. In the limit $m_{1,3} \to \infty$, the one loop result becomes exact and coincide with (half) the usual Casimir force expression for electrodynamics as obtained when the coupling to the boundaries is also very strong and Dirichlet boundary conditions are imposed.

Finally, measurements of fermions' anomalous magnetic moments are sensitive to the effects of new scalar fields coupled to matter. The anomalous magnetic moment is

$$a_f = \frac{g_f - 2}{2},$$ (130)

where $g_f$ is the fermion's g-factor. There are two effects to consider. First is the well-known result that at 1-loop the scalar particle corrects the QED vertex, modifying the anomalous magnetic moment by an amount [119,126,127]

$$\delta a_f \approx 2 \left( \frac{\beta(\phi) m_f}{4\pi m_{\rm Pl}} \right)^2,$$ (131)

where $m_f$ is the mass of the fermion. Second, the classical fifth force introduces systematic effects in the experiment, such as a modified cyclotron frequency, that must be accounted for in order to infer the correct measured value of $a_f$ [127,128].

In the case of the electron, the measurement of $a_e$ and the standard model prediction agree at the level of 1 part in $10^{12}$ [129]. Setting $\delta a_e \leq 10^{-12}$ yields the constraint [127]

$$\beta(\phi) \lesssim 10^{16}.$$ (132)

In the case of the chameleon where $\beta = m_{\rm Pl}/M$, this rules out $M < 80$ GeV.

In the case of the muon, the experimental measurement of the magnetic moment [130,131] and the standard model prediction [132–151] differ by 1 part in $10^9$ at 4.2 $\sigma$. A generic scalar field without a screening mechanism cannot account for this discrepancy without also being in tension with Solar System tests of gravity. However, it has recently been shown that both the chameleon and symmetron are able to resolve

this anomaly while also satisfying all other experimental bounds [128]. The chameleon parameters that accomplish this are

$$M = 500 \,\text{GeV and } \Lambda < \text{meV} . \tag{133}$$

Cosmologically, a chameleon with these parameters has an effective mass $m_{\text{cosmo}} > 10^{-13}$ eV and Compton wavelength $< 10^3$ km, so this theory does not significantly influence our universe on large scales.

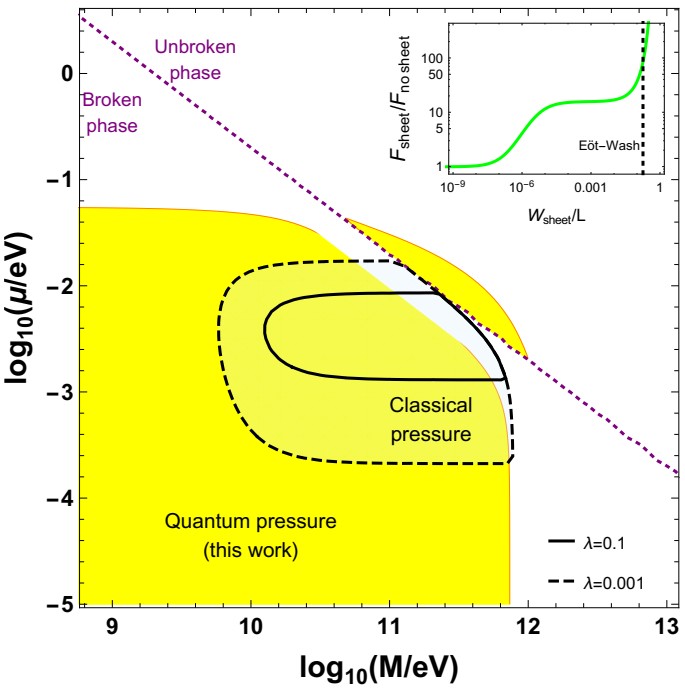

**Figure 3.** The constraints from the Eötwash experiment on a light symmetron as a function of the strong coupling scale $M$ for different values of the self-coupling $\lambda$. This involves the calculation with a five region geometry. The quantum constraints extend the classical bounds. In the insert, the enhancement of the quantum scalar interaction due to the presence of the thin plate between the external ones is represented. Reproduced from [125].

## 4. Astrophysical Constraints and Prospects

In this section, we discuss the ways in which screened fifth forces may be searched for using astrophysical objects beyond the solar system, specifically stars, galaxies, voids and galaxy clusters. We describe the tests that have already been conducted and the ways in which they may be strengthened in the future. Astrophysical constraints are most often phrased in terms of the $n = 1$ Hu–Sawicki model of $f(R)$ (taken as a paradigmatic chameleon-screened theory; [69,152,153]) and nDGP or a more general Galileon model (taken as paradigmatic Vainshtein-screened theories [59,73]).

Testing screening in astrophysics requires identifying unscreened objects where the fifth force should be manifest. Ideally this would be determined by solving the scalar's equation of motion given the distribution of mass in the universe, although the uncertainties in this distribution and the model-dependence of the calculation make more approximate methods expedient. This may be achieved by identifying proxies for the degree of screening in certain theories, which can be estimated from the observed galaxy field. In thin-shell screening mechanisms (chameleon, symmetron and the environmentally-dependent dilaton), it is the surface Newtonian potential of an object relative to the background scalar field value that determines whether it is screened (as discussed in Section 2). This screening criterion may be derived analytically for an object in isolation or in the linear cosmological regime (e.g., [48,49] for the chameleon), while N-body simulations in modified gravity

have shown that it is also approximately true in general when taking account of both environmental and self-screening [154–156] (see Figure 4). The threshold value of potential for screening is given by Equation 52: in $n = 1$ Hu–Sawicki $f(R)$, $\chi \simeq \frac{3}{2} f_{R0}$ so that probing weaker modified gravity (lower $f_{R0}$) requires testing objects in weaker-field environments [69]. Rigorous observational screening criteria are not so easy to derive in other screening mechanisms, although heuristically one would expect that in kinetic mechanisms governed by non-linearities in the first derivative of the scalar field, it is the first derivative of the Newtonian potential (i.e., acceleration) that is relevant, while in Vainshtein theories governed by the second derivative of the field, it is instead the space-time curvature (Section 2.5).

Several methods have been developed to build "screening maps" of the local universe to identify screened and unscreened objects. Shao et al. [157] apply an $f(R)$ scalar field solver to a constrained N-body simulation to estimate directly the scalar field strength as a function of position. Cabre et al. [154] use galaxy group and cluster catalogues to estimate the gravitational potential field and hence the scalar field by the equivalence described above. Desmond et al. [158] adopt a similar approach but include more contributions to the potential, model acceleration and curvature, and build a Monte Carlo pipeline for propagating uncertainties in the inputs to uncertainties in the gravitational field. By identifying weak-field regions, these algorithms open the door to tests of screening that depend on the local environment, with existing tests using one of the final two.

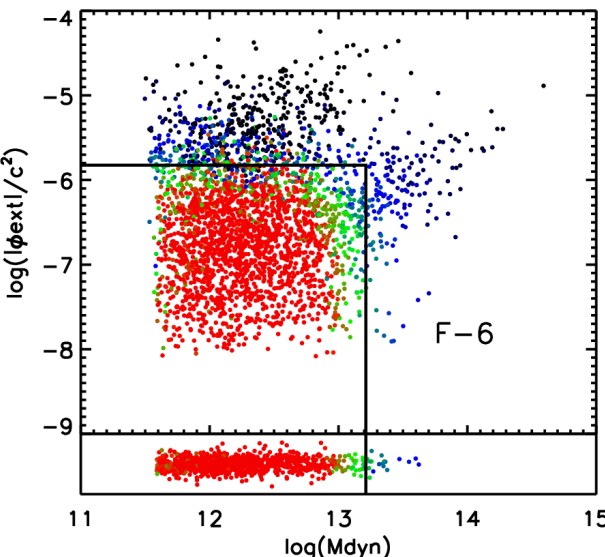

**Figure 4.** Halos produced in an $f(R)$ N-body simulation with $f_{R0} = 10^{-6}$. The x-axis is the total halo mass in $M_\odot$, the y-axis is the Newtonian potential sourced at the halo's position by mass within one Compton wavelength of the scalar field, and the points are colour-coded by the degree of screening with red fully unscreened and dark blue fully screened. The vertical and horizontal lines mark where the internal and external potentials equal $\frac{3}{2} f_{R0}$, showing that these cuts can reliably separate screened from unscreened galaxies. Reproduced from [154].

*4.1. Stellar Tests*

Gravitational physics affects stars through the hydrostatic equilibrium equation, which describes the pressure gradient necessary to prevent a star from collapsing under its own weight. In the Newtonian limit of GR, this is given by

$$\frac{dP}{dr} = -\frac{G_N M(r) \rho(r)}{r^2}. \tag{134}$$

In the presence of a thin-shell-screened fifth force, this becomes

$$\frac{dP}{dr} = -\frac{G_N M(r)\rho(r)}{r^2}\left[1 + 2\beta^2\left(1 - \frac{M(r_s)}{M(r)}\right)\Theta(r - r_s)\right],\tag{135}$$

with $\Theta(x)$ the Heaviside step function, $\beta$ the coupling coefficient of the scalar field and $r_s$ the screening radius of the star beyond which it is unscreened. In the case of chameleon theories, the factor $1 - \frac{M(r_s)}{M(r)}$ corresponds to the screening factor and is associated with the mass ratio of the thin shell, which couples to the scalar field. The stronger inward gravitational force due to modified gravity requires that the star burns fuel at a faster rate to support itself than it would in GR, making the star brighter and shorter-lived. The magnitude of this effect depends on the mass of the star: on the main sequence, low-mass stars have $L \propto G_N^4$ while high-mass stars have $L \propto G_N$ [159]. Thus in the case that the star is fully unscreened ($r_s = 0$), low-mass stars have $L$ boosted by a factor $(1 + 2\beta^2)^4$, and high-mass stars by $(1 + 2\beta^2)$.

To explore the full effect of a fifth force on the behaviour of stars, Equation (134) must be coupled with the equations describing stellar structure and energy generation. This has been achieved by modifying the stellar structure code MESA [159–162], enabling the heuristic expectations described above to be quantified (see Figure 5). The expectation that stars are brighter in modified gravity—and low-mass stars more so than high-mass—also leads to the prediction that unscreened galaxies would be more luminous and redder than otherwise identical screened ones. No quantitative test has been designed around this though because no galaxy formation simulation, including the effect of modified gravity on stars, has yet been run.

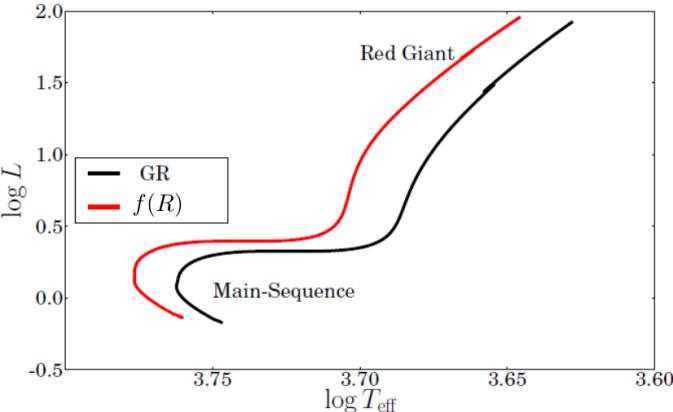

**Figure 5.** The colour-magnitude diagrams for a solar mass and metallicity star in GR (black) and Hu–Sawicki $f(R)$ gravity with $f_{R0} = 10^{-6}$ (red). $L$ is in units of solar luminosity and $T_{\mathrm{eff}}$ is in units of Kelvin.

Fifth forces also have important effects in astroseismology, the study of stellar oscillations. The equation of motion for small perturbations of mass elements in stars is

$$\ddot{\vec{\delta r}} = -\frac{1}{\rho}\frac{dP}{dr} + \vec{a},\tag{136}$$

with $\vec{a}$ the force per unit mass, which is $\vec{a} = -\vec{\nabla}\Phi$ in GR but $\vec{a} = -\vec{\nabla}\Phi - \frac{\beta}{m_{\mathrm{Pl}}}\vec{\nabla}\phi$ in the presence of a scalar field. Combining this equation with the other stellar structure equations gives the frequency of linear radial adiabatic oscillations

$$\omega^2 \sim \frac{G_N M}{R^3},\tag{137}$$

so that enhancing the effective value of $G$ due to the addition of a fifth force causes the pulsation period $\Pi$ to change according to

$$\frac{\Delta\Pi}{\Pi} = -\beta Q, \tag{138}$$

where $Q$ is the star's scalar charge.

Stellar oscillations are useful observationally because they provide several methods of determining distances to galaxies [163]. These afford a test of gravity when multiple distance indicators with different screening properties are combined. In particular, if a distance indicator is sensitive to $G_N$ and calibrated assuming GR, it will fail to give the correct distance to an unscreened galaxy in a fifth-force theory. This will lead to a discrepancy with the distance estimated using an indicator that is not sensitive to $G_N$, e.g., because it is based on the physics of a high-density, screened object.

This test has been carried out by comparing Cepheid and TRGB (Tip of the Red Giant Branch) distance indicators. Cepheids are post-main-sequence stars that oscillate radially by the $\kappa$-mechanism [164] when crossing the instability strip in the Hertzsprung–Russell diagram. The period of this pulsation is tightly correlated with the luminosity of the star, allowing Cepheids to be used as standard candles. TRGB stars are red giants that have become sufficiently hot for helium fusion to occur, moving the star onto the horizontal branch of the Hertzsprung–Russell diagram and leaving an observable discontinuity in the *I*-band magnitude. This occurs at an almost fixed absolute magnitude, making the TRGB feature another standard candle. The TRGB luminosity is sourced by a thin hydrogen-burning shell surrounding the helium-burning core, so if the core is screened then TRGBs exhibit regular GR behaviour. This occurs for $\chi \lesssim 10^{-6}$, which is the case for thin-shell theories that pass the tests described below. With Cepheids unscreened down to much lower values of $\chi$, this means that TRGB and Cepheid distances would be expected to disagree in unscreened galaxies. The fact that they seem not to—and that any discrepancy between them is uncorrelated with the galaxy environment—has yielded the constraint $f_{R0} \lesssim 10^{-7}$ [165,166]. Notice that astrophysical constraints yield tighter bounds on $f(R)$ models than solar system tests.

Variable stars are also useful for more general tests of gravity. [167] showed that the consistency between the mass estimates of Cepheids from stellar structure vs. astroseismology allows a constraint to be placed on the effective gravitational constant within the stars. Using just six Cepheids in the Large Magellanic Cloud afforded a 5% constraint on $G_N$, and application of this method to larger datasets spanning multiple galaxies will allow a test of the environment-dependence of $G_N$ predicted by screening. Screening may also provide a novel local resolution of the Hubble tension [166,168,169].

Finally, other types of stars are useful probes of the phenomenon of "Vainshtein breaking" whereby the Vainshtein mechanism may be ineffective inside astrophysical objects. An unscreened fifth force inside red dwarf stars would impact the minimum mass for hydrogen burning, and a constraint can be set by requiring that this minimum mass is below the lowest mass of any red dwarf observed [170,171]. It would also affect the radii of brown dwarf stars and the mass–radius relation and Chandresekhar mass of white dwarfs [172].

### 4.2. Galaxy and Void Tests

Screened fifth forces have interesting observable effects on the dynamics and morphology of galaxies. The most obvious effect is a boost to the rotation velocity and velocity dispersion beyond the screening radius due to the enhanced gravity. This is strongly degenerate with the uncertain distribution of dark matter in galaxies, although the characteristic upturn in the velocity at the screening radius helps to break this. In the case of chameleon screening, Naik et al. [173] fitted the rotation curves of 85 late-type galaxies with an $f(R)$ model, finding evidence for $f_{R0} \approx 10^{-7}$ assuming the dark matter follows an NFW profile but no evidence for a fifth force if it instead follows a cored profile as predicted by

some hydrodynamical simulations. This illustrates the fact that a fifth force in the galactic outskirts can make a cuspy matter distribution appear cored when reconstructed with Newtonian gravity, of potential relevance to the "cusp-core problem" [174] (see also [175]). Screening can also generate new correlations between dynamical variables; for example, Burrage et al. [176] use a symmetron model to reproduce the Radial Acceleration Relation linking the observed and baryonic accelerations in galaxies [177]. Further progress here requires a better understanding of the role of baryonic effects in shaping the dark matter distributions in galaxies, e.g., from cosmological hydrodynamical simulations in $\Lambda$CDM.

One way to break the degeneracy between a fifth force and the dark matter distribution is to look at the *relative* kinematics of galactic components that respond differently to screening. Since main-sequence stars have surface Newtonian potentials of $\sim 10^{-6}$, they are screened for viable thin-shell theories. Diffuse gas, on the other hand, may be unscreened in low-mass galaxies in low-density environments, causing it to feel the fifth force and hence rotate faster [178,179]:

$$\frac{v_g^2}{r} = \frac{G_N(1+2\beta^2)M(<r)}{r^2}, \quad \frac{v_*^2}{r} = \frac{G_N M(<r)}{r^2} \quad \Rightarrow \quad \frac{v_g}{v_*} = \sqrt{1+2\beta^2}, \tag{139}$$

where $M(<r)$ is the enclosed mass, and $v_g$ and $v_*$ are the gas and stellar velocities, respectively. We see that comparing stellar and gas kinematics at fixed galactocentric radius factors out the impact of dark matter, which is common to both. Comparing the kinematics of stellar Mg*b* absorption lines with that of gaseous H$\beta$ and [OIII] emission lines in six low-surface brightness galaxies, Vikram et al. [180] place the constraint $f_{R0} \lesssim 10^{-6}$. This result can likely be significantly strengthened by increasing the sample size using data from IFU surveys such as MaNGA or CALIFA—potentially combined with molecular gas kinematics, e.g., from ALMA—and by modelling the fifth force within the galaxies using a scalar field solver rather than an analytic approximation. A screened fifth force also generates asymmetries in galaxies' rotation curves when they fall nearly edge-on in the fifth-force field, although modelling this effect quantitatively is challenging so no concrete results have yet been achieved with it [181].

The strongest constraints to date on a thin-shell-screened fifth force with astrophysical range come from galaxy morphology. Consider an unscreened galaxy situated in a large-scale fifth-force field $\vec{a}_\phi$ sourced by surrounding structure. Since main-sequence stars self-screen, the galaxy's stellar component feels regular GR while the gas and dark matter also experience $\vec{a}_\phi$. This causes them to move ahead of the stellar component in that direction until an equilibrium is reached in which the restoring force on the stellar disk due to its offset from the halo centre exactly compensates for its insensitivity to $\vec{a}_\phi$ so that all parts of the galaxy have the same total acceleration [178,179]:

$$\frac{G_N M(<r_*)}{r_*^2} \, \hat{r}_* = 2\beta \, \vec{a}_\phi, \tag{140}$$

where $\vec{r}_*$ is the displacement of the stellar and gas centroids. This effect is illustrated in Figure 6a and can be measured by comparing galaxies' optical emission (tracing stars) to their HI emission (tracing neutral hydrogen gas). A second observable follows from the stellar and halo centres becoming displaced: the potential gradient this sets up across the stellar disk causes it to warp into a characteristic cup shape in the direction of $\vec{a}_\phi$. This is shown in Figure 6b. The shape of the warp can be calculated as a function of the fifth-force strength and range, the environment of the galaxy and the halo parameters that determine the restoring force:

$$z = \frac{2\beta \, a_\phi \, r^3}{G_N M(<r)}, \tag{141}$$

which can be simplified by assuming a halo density profile. Desmond et al. [182–184] create Bayesian forward models for the warps and gas–star offsets for several thousand galaxies observed in SDSS and ALFALFA, including Monte Carlo propagation of uncertainties in

the input quantities and marginalisation over an empirical noise model describing non-fifth-force contributions to the signals. This method yields the constraint $f_{R0} < 1.4 \times 10^{-8}$ at $1\sigma$ confidence, as well as tight constraints on the coupling coefficient of a thin-shell-screened fifth force with any range within 0.3-8 Mpc [185] (see Figure 7a). Subsequent work has verified using hydrodynamical simulations that the baryonic noise model used in these analyses is accurate [186]. The value of $10^{-8}$ is around the lowest Newtonian potential probed by any astrophysical object, so it will be very hard to reach lower values of $f_{R0}$. Lower coupling coefficients may, however, be probed using increased sample sizes from upcoming surveys such as WFIRST, LSST and SKA, coupled with estimates of the environmental screening field out to higher redshift using deeper wide photometric surveys.

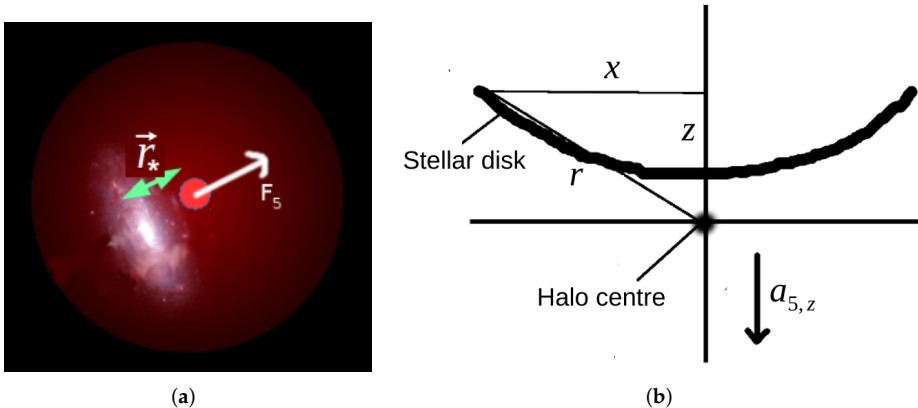

(a)          (b)

**Figure 6.** Cartoons illustrating (**a**) separation of stars and gas in galaxies under a fifth force, and (**b**) the warping of stellar disks. (**b**) is reproduced from [184].

The above tests target thin-shell-screened fifth forces. The Vainshtein mechanism is harder to probe due to the efficiency of its screening on small scales and the difficulty of developing robust observational proxies for objects' degrees of screening. While LLR is sensitive to cubic Galileons with small crossover scale $r_c \sim \mathcal{O}(100)$ kpc [37], the larger values $r_c \sim 6000$ Mpc required for self-acceleration [187] must be probed on galactic or cosmological scales. The most promising method for this utilises the breaking of the Strong Equivalence Principle (SEP) that Galileons imply [188] in the presence of black holes. Galileons couple to the trace of the stress-energy tensor, which is equivalent to density but excludes gravitational binding energy. This means that non-relativistic objects (e.g., stars, gas and dark matter in galaxies) have a scalar charge-to-mass ratio equal to the coupling coefficient $\beta$, while black holes are purely relativistic objects with $Q = 0$. Thus, in the presence of an unscreened large-scale Galileon field, the supermassive black holes at galaxies' centres will lag behind the rest of the galaxy, which is measurable by comparing the galaxies' light with radio or X-ray emission from the Active Galactic Nuclei (AGN) powered by the black hole. Two situations can lead to an unscreened Galileon field. The first is in galaxy cluster environments: an extended distribution of mass does not Vainshtein-screen perfectly in its interior [189], so a residual fifth-force field is present in cluster outskirts. This leads to $\mathcal{O}(\text{kpc})$ offsets between black holes and satellite galaxy centres for realistic cluster parameters. Sakstein et al. [190] solve the Galileon equation of motion for a model of the Virgo cluster and use the fact that the black hole in the satellite galaxy M87 is within 0.03 arcsec of the galaxy centre to rule out $\mathcal{O}(1)$ coupling coefficients for $r_c \lesssim 1$ Gpc. Second, the Galileon symmetry implies that the linear contribution to the field on cosmological scales is unscreened [191,192], allowing black hole offsets to develop

even for field galaxies. Assuming a constant density $\rho_0$ in the centre of the halo, the black hole offset in this case is given by [188]

$$R = 0.1 \, \text{kpc} \left(2\beta^2\right) \left( \frac{|\nabla \Phi_N^{\text{ext}}|}{20 \, (\text{km/s})^2/\text{kpc}} \right) \left( \frac{0.01 M_\odot/\text{pc}^3}{\rho_0} \right), \tag{142}$$

where $\nabla \Phi_N^{\text{ext}}$ is the unscreened large-scale gravitational field, proportional to the Galileon fifth-force field. Bartlett et al. [193] modelled this field using constrained N-body simulations of the local ∼200 Mpc and forward-modelled the offsets in 1916 galaxies with AGN, including a more sophisticated model for the halo density profiles, to set the bound $\beta < 0.28$ for $r_c \gtrsim 1/H_0$ (see Figure 7b). This probes the cosmologically-relevant region of the Galileon parameter space, complementing cosmological probes such as the Integrated Sachs Wolfe (ISW) effect (see Section 5). It could be improved to probe smaller $r_c$ values by modelling the full, non-linear dynamics of the Galileon within the test galaxies. Another possible signature is "wandering" black holes seemingly unassociated with a galaxy [190].

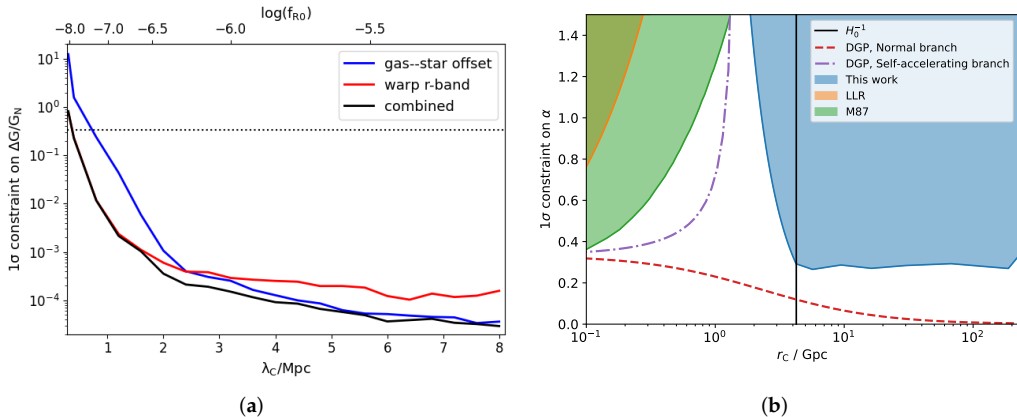

**Figure 7.** (**a**) Constraints on a thin-shell-screened fifth force from displacement between the centres of emission of optical and HI light in galaxies (i.e., separation of stars and gas), and stellar warps observed in the *r*-band. $\lambda_C$ is the Compton wavelength of the scalar, $\Delta G/G_N \equiv 2\beta^2$ and the horizontal dashed line marks $\Delta G/G_N = 1/3$ in $f(R)$ gravity. Reproduced from [185]. (**b**) Constraints on the Galileon coupling coefficient (on the plot denotes by $\alpha$) as a function of the theory's crossover scale. The orange region is excluded by Lunar Laser Ranging, the green region from the position of the supermassive black hole in M87, and the blue region from a statistical analysis of the black hole positions in field galaxies. Reproduced from [193].

While galaxies are the directly observable tracers of the cosmic web, much dynamical information can be found in *voids*, the underdense regions that comprise most of the universe's volume. These are particularly promising for testing screening because they are the regions where it is least efficient. Their usefulness is, however, hampered by the ambiguity that exists in defining voids and by the fact that voids must be identified observationally using biased tracers of the underlying density field (galaxies). Voids in modified gravity have been studied through both analytic [194,195] and simulation [196,197] methods. Typically, the enhanced growth of structure in the presence of a fifth force causes voids to become larger and emptier. In addition, when voids are identified through lensing, the modified relation between mass and lensing potential can affect the lensing signal [79,198]. Voids can also be cross-correlated with galaxies to infer the growth rate of structure [199], used in the ISW effect [200], integrated along the line of sight to produce projected 2D voids [201], and used as a means of splitting samples of galaxies into high-density (screened) and low-density (unscreened) environments or in marked correlation functions [202,203]. Finally, the redshift-space anisotropy of voids is a powerful probe of the nature of gravity through

redshift space distortions [204]. Future surveys will improve 3D spectroscopic void finding and the calibration of photometric void finders with robust photometric redshifts.

### 4.3. Galaxy Cluster Tests

A fifth force causes a structure to grow more quickly, leading to more cluster-sized halos at late times. This is, however, counteracted by screening and the Yukawa cut off due to the mass of the scalar field so that cluster abundance only deviates significantly from the ΛCDM expectation at lower masses and in sparser environments [205]. The excursion set formalism for halo abundance provides a good description under chameleon gravity as well [206], albeit with a modified collapse threshold $\delta_c$, and has been used to constrain $f_{R0} \lesssim 10^{-5}$ in the Hu–Sawicki model [207,208]. Similar constraints are achievable using the peaks in the weak lensing convergence field, which trace massive halos [209]. Other formalisms for calculating cluster abundance in the presence of a fifth force have also been developed [210–213]. Qualitatively similar results hold for Vainshtein-screened theories, where, although the centres of clusters are efficiently screened, massive halos grow at an increased rate because of enhanced accretion due to the fifth force in the surrounding matter [214]. This can be significantly altered for K-mouflage models where clusters are not screened so we expect massive halos to be more abundant than in ΛCDM. This is illustrated in Figure 8; the "arctan" models are particularly interesting because they pass the solar system tests.

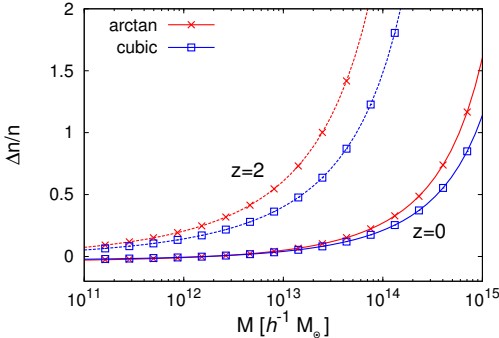

**Figure 8.** Fractional increase in halo abundance in K-mouflage relative to ΛCDM as a function of halo mass. Results are shown for two different redshifts and K-mouflage models. Reproduced from [56].

The internal structures of cluster halos are also altered by modified gravity, particularly through an increase in the concentration of the Navarro–Frenk–White profile [215–217], although this is hard to use to set constraints due to degeneracies with the impact of baryons. Another important effect is on the boundary of cluster halos, namely the splashback radius where accreting dark matter turns around after the first infall [218]. This is marked by a sharp drop in the logarithmic density slope and, consequently, in the lensing signal and subhalo profile. Adhikari et al. [219] studied the splashback feature in both chameleon and symmetron models, finding that for viable and interesting values of the fifth-force properties, the splashback radius is increased relative to GR in Vainshtein models and reduced in chameleon. This results from competition between the enhanced acceleration of accreting matter and reduced dynamical friction within the halos. There is, however, controversy observationally about the location of the cluster splashback radius [220–222], so these predictions cannot be used to set concrete constraints. Further out, the halo–matter correlation function is enhanced by a fifth force [207,223].

A powerful and general method for probing modified gravity leverages the inequality between the two weak-field metric potentials, a violation of the weak equivalence principle. This leads to a difference between the dynamical and lensing masses of objects: while photons respond to the sum of the metric potentials, non-relativistic tracers are affected solely by the time–time potential. Thin-shell screening alters the Newtonian potential but

not the lensing one, which in the parametrised post-Newtonian framework is captured by the parameter $\gamma$. Although $\gamma$ may be constrained on $\mathcal{O}(\text{kpc})$ scales by comparing strong lensing and stellar dynamical mass estimates [224,225], it has found the most use on the scale of clusters. An approximation for chameleon theories of the Jordan–Brans–Dicke type is [226]

$$M_{\text{dyn}}(r) \simeq \left(1 + \frac{\Theta(r - r_{\text{c}})}{3 + 2\omega_{\text{BD}}}\left[1 - \frac{M(r_{\text{c}})}{M(r)}\right]\right) M(r)_{\text{lens}}, \tag{143}$$

where $\Theta$ is the Heaviside step function, $\omega_{\text{BD}}$ is the JBD parameter (see Section 2.7.5) and the radius at which the scalar field transitions to its background value is given by

$$r_{\text{c}} \simeq \frac{32\pi G\rho(r_{\text{s}})r_{\text{s}}^3}{3 + 2\omega_{\text{BD}}}\frac{1}{1 - A^{-2}(\phi_{\text{env}})} - r_{\text{s}}. \tag{144}$$

Here $\phi_{\text{env}}$ is the cosmological boundary condition for the field far from the cluster (e.g., $1 - A^{-2}(\phi) \simeq f_{R0}$ in the $f(R)$ case) and $r_s$ is the scale length of the cluster's assumed-NFW density profile. The difference between dynamical and "true" masses of clusters in $f(R)$ gravity has also been calibrated from N-body simulations in [227]:

$$\frac{M_{\text{dyn}}}{M_{\text{lens}}} = \frac{7}{6} - \frac{1}{6}\tanh\left[p_1\left(\log_{10}(M_{\text{lens}}/M_{\odot}) - p_2\right)\right], \tag{145}$$

where $p_1 = 2.21$ and $p_2 = 1.503\log_{10}\left[\frac{|f_R(z)-1|}{1+z}\right] + 21.64$. This works well for $f_{R0} \in [10^{-6.5}, 10^{-4}]$ and $z \in [0, 1]$. To test this effect, strong cluster lensing may be compared to X-ray masses or the velocity dispersions of the cluster galaxies [189,228], and stacked weak lensing can be compared to Sunyaev–Zel'dovich masses or infall motions at the cluster outskirts [229]. Dynamical masses can also be estimated from X-ray data of cluster temperature and pressure profiles. The combination of weak lensing measurements with measurements of the X-ray brightness, temperature and Sunyaev–Zel'dovich signal from the Coma cluster [230] (or from multiple clusters' weak lensing and X-ray signals [231]) implies $f_{R0} \lesssim 6 \times 10^{-5}$, and this test has also been applied to Galileons [232]. The modification to clusters' dynamical masses under a fifth force can be probed without requiring lensing data by assuming that the gas fractions of clusters are constant in order to estimate the true total mass. This is capable of constraining $f(R)$ to the $f_{R0} \sim 5 \times 10^{-5}$ level [233]. All of these tests will benefit from enlarged cluster samples in the future.

## 5. Cosmological Consequences

### 5.1. Screening and Cosmic Acceleration

Screened fifth forces coupled to matter also have interesting cosmological consequences. In the modified gravity models studied above, the screening mechanisms are necessary to make the models consistent with observations at small scales. As detailed in Sections 2.3 and 2.4, we can classify the screening types into non-derivative and derivative screening mechanisms. From the former, the chameleon is the most popular example, appearing in popular models such as Hu–Sawicki $f(R)$. For the latter, the Vainshtein and K-mouflage mechanisms are the characteristic ones, appearing in subsets of Horndeski theory, such as models with a modified kinetic term (for K-mouflage) or models such as Cubic Galileons, which feature the Vainshtein screening as a way to evade small scale gravitational constraints.

No-go theorems [35,87,88] were developed for chameleon-screened theories, and they state namely that (i) the Compton wavelength of such scalars can be at most $\simeq 1$ Mpc at the present cosmic density, which means that the effective range of these theories is restricted to non-linear scales in large scale structure formation and they have no effect on the linear growth of structures; and (ii) that the conformal factor (64) relating the Einstein and Jordan frames of these theories is essentially constant in one Hubble time; therefore, these scalar fields cannot be responsible for self-acceleration and one needs to invoke either

a cosmological constant term or another form of dark energy to explain the acceleration of the expansion of the Universe. More precisely, in the context of chameleon-screened models one can show that the equation-of-state of dark energy at late times is of order [53]

$$\omega_\phi + 1 \simeq \mathcal{O}(\frac{H^2}{m^2}) \tag{146}$$

where $m$ is the mass of the light scalar. The bound from solar systems on the mass ratio $m/H \gtrsim 10^3$ coming from solar system tests, see (80), implies that the equation-of-state is indistinguishable from the one of a cosmological constant. On the other hand, these theories have effects on large scale structures and then irrespective of what would drive the acceleration one could test the screening effects at the cosmological perturbation level.

In the second class of models, the scalar field evolves significantly on cosmic timescales, as in the case of cubic Galileons, kinetic gravity braiding models and K-mouflage models. These models present either K-mouflage or Vainshtein screenings and, therefore, are not affected by the no-go theorems.

In the following sections, we will present the different ways in which these screened modified gravity theories affect cosmological observables and the current and future bounds that can be placed on their parameters.

*5.2. Screening and Structure Formation*

The formation of a large scale structure is affected by the presence of modified gravity. Screening could play a role too as we will see below as the growth of structure depends on the type of screening mechanisms. For derivative screening, the growth is affected at the linear level in a scale-independent way. For non-derivative screenings, the linear growth is modified in a scale-dependent way. The latter can be easily understood as there is a characteristic length scale, i.e., the Compton wavelength of the scalar field, beyond which modified gravity is Yukawa-suppressed. Non-linear effects are also important and tend to dampen the effects of modifying gravity on small scales.

As an example and on cosmological scales, the $f(R)$ modification of the Einstein–Hilbert action leads to a modified Poisson equation, which can be expressed as

$$\nabla^2 \Phi = \frac{16\pi G}{3} a^2 \delta\rho - \frac{a^2}{6} \delta R \, , \tag{147}$$

in comoving coordinates and the term $\delta\rho$ is the matter density fluctuation compared to the cosmological background and $\Phi$ the modified Newtonian potential. Furthermore, the fluctuation of the Ricci scalar, $\delta R = R - \bar{R}$ compared to the cosmological background $\bar{R}$ and is expressed as

$$\nabla^2 \delta f_R = \frac{a^2}{3} [\delta R - 8\pi G \delta\rho] \, . \tag{148}$$

The variation of the function $f(R)$ is given by $\delta f_R = f_R(R) - f_R(\bar{R})$. In these equations, we have assumed a quasi-static approximation. It can be shown [234] that despite the fact that these equations are non-linear in $\delta R$, they are self-averaging. This means that on large scales one recovers $\delta R \to 0$. Using these governing equations, one can solve perturbatively the Vlasov–Poisson system of equations, which consists in the first approximation (no vorticity and single-stream regime) of the continuity, Euler and Poisson equations, in powers of the linear growth factor. The results of these computations at 1-loop order and beyond can be seen in References [234–239].

In scalar-tensor theories with screening and a conformal factor $A(\phi)$, particles feel a total gravitational potential $\Phi$, which is the sum of the standard Newtonian term $\Phi_N$ and an additional contribution $\Phi_A$,

$$\Phi = \Phi_N + \Phi_A \, , \tag{149}$$

where the governing equations are given by

$$\frac{1}{a^2}\Delta\Phi_N = 4\pi G\delta\rho, \qquad \Phi_A = \ln\frac{A}{\bar{A}} \simeq (A - \bar{A}) \tag{150}$$

where it is assumed that $A(\phi) \simeq 1$ to satisfy constraints on the variation of fermionic masses. As a result, $\ln A \simeq A - 1$ and the dependence on $\ln A$ of the Newtonian potential $\Phi$ becomes linear in $A$. This additional gravitational potential implies that matter particles of mass $m$ are sensitive to a "fifth force" given by

$$\vec{F}_\phi = -m\vec{\nabla}\ln A. \tag{151}$$

This fifth force is the one that leads to a modification of the growth of structures.

### 5.3. Cosmological Probes: CMB and Large Scale Structure

Historically, the background expansion of the Universe has been the traditional way of testing cosmological models, and this has been developed mostly through the study of standard candles, especially with the use of observations of supernovae SNIa [3,240]. However, recent constraints on the equation-of-state parameter of dark energy are overall consistent with a cosmological constant $w \approx -1$ [241]. This, plus the fact that self-acceleration is mostly ruled out in the most popular screened scalar field models, has led to the tendency in the literature to look for features of dark energy and modified gravity in the formation of structures and the modification of gravitational lensing. Moreover, other interesting tensions in the data, such as the $H_0$ tension [242], cannot be satisfactorily resolved with late-time dynamics of a dark energy field, according to the latest analysis [243,244] and therefore will not be covered in this section. Therefore, in the following section, we will concentrate mostly on the integrated Sachs Wolfe effect in the CMB, lensing of the CMB and the formation of structures probed by the Galaxy power spectrum and its effect on weak lensing (cosmic shear).

### 5.3.1. ISW and CMB Lensing

The relic radiation from the early Universe that we observe in the GHz frequency range, called the Cosmic Microwave Background, is one of the most powerful cosmological probes. It constrains not only the background of the Universe but also its growth of structure. Its primary anisotropies, imprinted at the time of recombination, provide plenty of information about the constituents of the Universe; while its secondary anisotropies, which happen later when the CMB photons are traversing the Universe, provide information about the intervening medium, the expansion of the Universe and the large scale structures. For studying late modified gravity and dark energy, these secondary anisotropies are the most important probes, namely the Integragted Sachs–Wolfe effect (ISW) ([245–247] that affect the power spectrum at low multipoles (large scales) and lensing of the CMB [248,249] that affects the spectrum at small scales (high multipoles).

In the case of ISW, the effect is observed as a temperature fluctuation caused by time variations in the gravitational potentials that are felt by photons when they enter and leave potential wells (or potential hills) when entering dark matter halos (or voids). The effect on the CMB temperature $T$ is given by

$$\frac{\delta T}{T}(\hat{n}) = -\int_{\eta_0}^{\eta_*} d\eta \frac{\partial(\Psi + \Phi)}{\partial\eta}, \tag{152}$$

where $\eta_*$ is the conformal time at the last scattering surface and $\eta_0$ at the observer. By changing the time evolution of the gravitational potentials, MG models affect the large scales of the CMB power spectrum through the ISW effect. The ISW effect played a major role in ruling out cubic Galileon models, which are the only non-trivial parts left from the Horndeski theory after GW170817. In [214], cubic Galileons were analysed, and it was found that in the presence of massive neutrinos (model dubbed $\nu$Galileon, in red in

Figure 9), the models were still a very good fit to CMB temperature, lensing and Baryon Acoustic Oscillation (BAO) data, using Planck-2013 temperature and lensing [250] and WMAP-9 polarisation [251] data. For BAO they used 6dF, SDSS DR7 and BOSS DR9 data ([252–254]). In the absence of massive neutrinos (model dubbed Galileon in Figure 9), however, ΛCDM was favoured by the data. Nevertheless, they showed that the νGalileon model shows a negative ISW effect that is hard to reconcile with current observations. More recently, a paper by [255] performed a detailed study of self-accelerating Galileon models using CMB data from Planck-15 in temperature and polarisation and CMB lensing [256]. They also included BAO data, $H_0$ data and ISW data. As in the older analysis, they showed that the cubic Galileon predicts a negative ISW effect and, therefore, it is in a $7.8\sigma$ tension with observations, effectively ruling this model out. Furthermore, in [257], the effect of different neutrino masses and hierarchies was analysed, and it was also found out that all cubic, quartic and quintic Galileons remain ruled out by CMB and ISW observations.

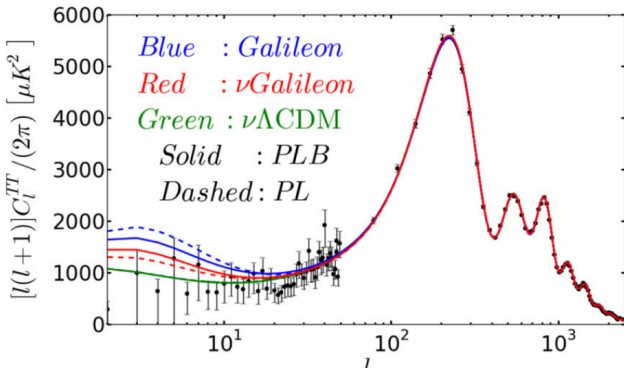

**Figure 9.** CMB temperature power spectrum. In black dots, data from Planck-2013; in blue, the cubic Galileon model without massive neutrinos; in red, the same model in the presence of massive neutrinos; and in green, baseline ΛCDM with standard neutrino mass. The difference between solid and dashed lines corresponds to an analysis of Planck with and without BAO data, respectively. Reproduced from [214] thankfully provided by Alex Barreira.

5.3.2. Cosmological Perturbations in Large Scale Structure

As mentioned above in the corresponding sections for $f(R)$ and scalar field models, the dynamics of the field at large scales is given by the Poisson equation and the corresponding Klein–Gordon equation. However, when including the full energy-momentum tensor, the first-order perturbed Einstein equations in Fourier space give two equations that describe the evolution of the two gravitational potentials $\Phi$ and $\Psi$. In the quasistatic approximation, these equations read

$$-k^2\Phi(a,k) = 4\pi G a^2 \mu(a,k)\rho(a)\Delta(a,k) \; ; \tag{153}$$
$$-k^2\Phi_K(a,k) = 4\pi G a^2 \Sigma(a,k)\rho(a)\Delta(a,k) \; . \tag{154}$$

where $\rho(a)$ is the average dark matter density and $\Delta(a,k) = \delta + 3aH\theta$ is the comoving density contrast with $\delta$, the fractional overdensity, and $\theta$ the peculiar velocity. We have denoted by

$$\Phi_L(a,k) = \frac{\Phi(a,k) + \Psi(a,k)}{2} \tag{155}$$

the lensing potential. The ratio of the two gravitational potentials is denoted as $\eta$, gravitational anisotropic stress or gravitational slip

$$\eta(a,k) \equiv \Psi(a,k)/\Phi(a,k) \; . \tag{156}$$

The scale and time-dependent functions $\eta(a,k)$, $\mu(a,k)$ and $\Sigma(a,k)$ stand for all possible deviations of Einstein gravity in these equations, being equal to unity when standard GR is

recovered and can encompass any modification by a scalar-tensor theory at the linear level in perturbations. Given that there are only two scalar degrees of freedom, it means that of course there is a relationship between $\mu$, $\Sigma$ and $\eta$ and they are related by

$$\Sigma(a,k) = \frac{\mu(a,k)}{2}[1 + \eta(a,k)] \ . \tag{157}$$

The $\mu(a,k)$ function is usually probed best by galaxy clustering experiments that directly trace the evolution of the $\Phi$ potential, since this one affects non-relativistic particles. $\mu$ is directly related to the effective Newtonian constant defined above in (9) as

$$G_{eff}/G_N = \mu \tag{158}$$

in the linear regime and in Fourier space. On the other hand, relativistic particles, and therefore light, follows the equation for $\Phi(a,k) + \Psi(a,k)$, meaning that gravitational weak lensing is mostly sensitive to the function $\Sigma(a,k)$.

$f(R)$ Models and Chameleon Theories

For the $f(R)$ theories described above, these expressions reflect the presence of an extra fifth force. In particular, it is convenient to introduce the mass of the scalaron field, i.e., the scalar field associated with the $f(R)$ models [53]

$$m_{f_R}^2 = \frac{f_R}{3 f_{RR}} \sim \frac{1}{3 f_{RR}} \tag{159}$$

where we have used that $R \simeq \rho/m_{\rm Pl}^2$ at a late time in the Universe. Neglecting the anisotropic stress, the expressions for $\mu$ and $\eta$ read [258]

$$\mu(a,k) = \frac{1}{f_R(a)} \frac{4 + 3k^2 a^{-2} m_{f_R}^{-2}(a)}{3(1 + k^2 a^{-2} m_{f_R}^{-2}(a))} \ , \tag{160}$$

$$\Sigma(a) = \frac{1}{f_R(a)} \ , \tag{161}$$

Given the constraints on $f_{R,0}$ mentioned above, the modifications of lensing are practically non-existent and $\Sigma(a,k) \simeq 1$ with great precision.

It is convenient to rewrite the above expressions as

$$\mu(a,k) \quad = A^2(a)\left(1 + \frac{2\beta(a)^2}{1 + \frac{m^2(a)a^2}{k^2}}\right), \tag{162}$$

$$\Sigma(a,k) \quad = A^2(a), \tag{163}$$

where in the case of $f(R)$ models, we have $\beta(a) = \beta = 1/\sqrt{6}$ and $m_{f_R}(a) = m(a)$, where $\beta(a)$ is the coupling at the minimum of the effective potential $V_{\rm eff}(\phi) = V(\phi) + (A(\phi) - 1)\rho$ as a function of $a$ with $\rho \propto a^{-3}$ and similarly for the mass of the scalar field $m(a)$. These expressions are valid for any chameleon theories.

In all chameleon theories, there is a one-to-one correspondence between the coupling and mass variations as a function of the scale factor and the potential $V(\phi)$ and coupling function $A(\phi)$, which is called the tomographic map. This allows to parameterise the chameleon models with the function $m(a)$ and $\beta(a)$. The mapping reads [88]

$$\frac{\phi(a)}{m_{\rm Pl}} = \frac{\phi_{\rm ini}}{m_{\rm Pl}} + 9 \int_{a_{\rm ini}}^{a} dx \frac{\beta(x)\Omega_m(x)H^2(x)}{x m^2(x)} \tag{164}$$

where $\Omega_m$ is the matter fraction of the Universe. In this expression, the matter fraction and the Hubble rate can be taken as the ones of the standard model as solar system tests

imply that chameleon models essentially coincide with $\Lambda$CDM at the background level. The potential itself is given by

$$V(a) = V_{\text{ini}} - \frac{3}{m_{\text{Pl}}^2} \int_{a_{\text{ini}}}^{a} dx \frac{\beta^2(x)\rho^2(x)}{x^2 m^2(x)}. \tag{165}$$

This provides a parametric reconstruction of $V(\phi)$. For the Hu–Sawicki models of $f(R)$, we have [53]

$$m_{f_{\text{R}}}(a) = m_0 \left( \frac{\frac{\Omega_{m0}}{a^3} + 4\Omega_{\Lambda 0}}{\Omega_{m0} + 4\Omega_{\Lambda 0}} \right)^{(n+2)/2} \tag{166}$$

where $\Omega_\Lambda$ is the dark energy fraction and $\Omega_{m0}$ the matter fraction now. The mass of the scalaron now is given by

$$m_0 = \frac{H_0}{\sqrt{(n+1)|f_{R0}|}} (4\Omega_{\Lambda 0} + \Omega_{m0})^{1/2}. \tag{167}$$

which is greater than $H_0$ for small $|f_{R0}| \ll 1$.

Finally, the $\mu$ parameterisation allows one to see how screening works on cosmological linear scales [50]. Defining the comoving Compton wavelength

$$\lambda_c(a) = \frac{1}{am(a)} \tag{168}$$

we find that for scales outside the Compton wavelength, i.e., $k \lesssim \lambda_c$ we have

$$\mu(a,k) \simeq A^2(a) \simeq 1 \tag{169}$$

and GR is retrieved. This corresponds to the Yukawa suppression of the fifth force induced by the light scalar. On the contrary, when $k \gtrsim \lambda_c$, we have an enhancement of the gravitational interaction as

$$\mu(a,k) \simeq 1 + 2\beta^2(a) \tag{170}$$

which is simply due to the exchange of the nearly-massless scalar field between overdensities.

As a result, we can have a qualitative description of chameleon models such as $f(R)$ on the growth of structures [259]. First of all, on very large scales, GR is retrieved and no deviation from $\Lambda$-CDM is expected. On intermediate scales, deviations are present as (170) is relevant. Finally, on much smaller scales, the screening mechanism prevents any deviations and GR is again retrieved. The onset of the modified gravity regime is set by the mass of the scalar now, which is constrained by the solar system tests to be in the sub-Mpc range. This falls at the onset of the non-linear regime of growth formation, and therefore, one expects the effects of modified gravity to be intertwined with non-linear effects in the growth process.

Jordan–Brans–Dicke Models

For the JBD models with a mass term, these functions are given by [260,261]

$$\mu(a,k) = \frac{1}{\bar{\phi}} \frac{2(2 + \omega_{BD})}{3 + 2\omega_{BD}}, \tag{171}$$

$$\eta(a,k) = \frac{2 + \omega_{BD}}{1 + \omega_{BD}}, \tag{172}$$

so that for cosmological purposes

$$\Sigma(a) = 1. \tag{173}$$

In this case, lensing is not affected at all.

Horndeski Theory

For a generic Horndeski theory (of second-order in the equations of motion), these two functions $\mu$ and $\eta$ can be expressed as a combination of five free functions of time $p_{1,2,3,4,5}$, which are related to the free functions $G_i$ in the Horndeski action [258,260]

$$\mu(a,k) = \frac{p_1(a) + p_2(a)k^2}{1 + p_3(a)k^2}, \tag{174}$$

$$\eta(a,k) = \frac{1 + p_3(a)k^2}{p_4(a) + p_5(a)k^2}. \tag{175}$$

There is another physically more meaningful parametrisation of the linear Horndeski action, given by [262], which is related to the effective field theory of dark energy [32,263,264], where small deviations to the background cosmology are parameterised linearly. This parametrisation is of great help when discussing current cosmological constraints. It is defined using four functions of time $\alpha_M$, $\alpha_K$, $\alpha_B$ and $\alpha_T$ plus the effective Planck mass $M_\star^2$ and a function of time for a given background specified by the time variation of the Hubble rate $H(a)$ as a function of the scale factor $a$. The term $\alpha_T$ measures the excess of speed of gravitational waves compared to light, and therefore, as we previously mentioned, after the event GW170817, this term is constrained to be effectively zero. The term $\alpha_K$ quantifies the kineticity of the scalar field and therefore appears in models such as K-mouflage, which require the K-mouflage screening. The coefficient $\alpha_B$ quantifies the braiding or mixing of the kinetic terms of the scalar field and the metric and can cause dark energy clustering. It appears in all modified gravity models where a fifth force is present [265]. It receives contributions also from terms related to the cubic Galileons, which present the Vainshtein screening. Finally, $\alpha_M$ quantifies the running rate of the effective Planck mass, and it is generated by a non-minimal coupling. This parameter modifies the lensing terms, since it directly affects the lensing potential. It appears in $f(R)$ models, where the chameleon screening is necessary, as we have seen.

DGP Models

Cosmological linear perturbations for DGP have been worked out in [266]. In the paper by [260], it is assumed that the small-scale (quasi-static) approximation is valid, i.e., $k/a \gg r_5 \mathcal{H}$ and obtains

$$-k^2 \Psi = 4\pi G_N \left(1 - \frac{1}{3\gamma}\right) \bar{\rho} a^2 \delta, \tag{176}$$

and

$$-k^2 \Phi = 4\pi G_N \left(1 + \frac{1}{3\gamma}\right) \bar{\rho} a^2 \delta, \tag{177}$$

where $\gamma = 1 + 2\epsilon H r_5 w_{\text{eff}}$. This corresponds to

$$\mu(a) = 1 + \frac{1}{3\gamma}, \tag{178}$$

and for all practical purposes, we can set $\Sigma = 1$ within the cosmological horizon (see [260]).

*5.4. Large Scale Structure Observations: Galaxy Clustering and Weak Lensing*

The most important probes for a large scale structure, especially in the upcoming decade with the advent of new observations by DESI [267][12], Euclid [268,269][13], Vera Rubin [270][14] and WFIRST [271][15], will be galaxy clustering and weak lensing. Galaxy clustering measures the 2-point-correlation function of galaxy positions either in three dimensions, i.e., angular positions and redshift, or in effectively two dimensions (angular galaxy clustering) when the redshift information is not particularly good. In Fourier space, this correlation function of galaxies, known as the observed galaxy power spectrum $P_{gg}^{obs}$

is directly related to the power spectrum of matter density perturbations $P_{\delta\delta,zs}$ in redshift space by

$$P_{\text{gg}}^{\text{obs}}(z,k,\mu_\theta) = \text{AP}(z)P_{\delta\delta,\text{zs}}(k,z)E_{\text{err}}(z,k) + P_{\text{shot}}(z)\,, \tag{179}$$

where $AP(z)$ corresponds to the Alcock–Paczynski effect, $E_{err}(z,k)$ is a damping term given by redshift errors and $P_{\text{shot}}(z)$ is the shot noise from estimating a continuous distribution out of a discrete set of points. $\mu_\theta$ is the cosine of the angle between the line of sight and the wave vector **k**. Furthermore, the redshift space power spectrum is given by

$$P_{\delta\delta,\text{zs}}(z,k,\mu_\theta) = \text{FoG}(z,k,\mu_\theta)K^2(z,\mu_\theta;b(z);f(z))P_{\delta\delta}(k,\mu_\theta,z)\,, \tag{180}$$

where $\text{FoG}(z,k,\mu_\theta)$ is the "Fingers of God" term that accounts for non-linear peculiar velocity dispersions of the galaxies, and K is the redshift space distortion term that depends—in linear theory, where it is known as the Kaiser term [272]—on the growth rate $f(z)$ and the bias $b(z)$, but can be more complicated when taking into account non-linear perturbation theory at mildly non-linear scales. For a detailed explanation of these terms, we refer the reader to [273] and the many references therein.

Relativistic effects in galaxy clustering may provide a particularly sensitive probe of fifth forces and screening. With relativistic effects included, the cross-correlation of two galaxy populations with different screening properties yields a dipole and octopole in the correlation function due to the effective violation of the weak equivalence principle—as encapsulated in Euler's equation—as the galaxies in the two groups respond differently to an external fifth-force field [274,275]. This may be observable in upcoming spectroscopic surveys such as DESI [276]. Reference [277] showed that the octopole is a particularly clean probe of screening per se (as opposed to the background modification that screened theories also imply) because it is not degenerated with the difference in bias between the galaxy sub-populations.

The second probe, weak lensing, is the 2-point correlation function of cosmic shear, which emerges when galaxy shapes become distorted, their ellipticities increased and their magnitudes changed, due to light travelling through large scale structures in the Universe, from the source to the observer [278]. These ellipticities and magnitudes are correlated through the distribution of matter in the Universe and the expansion. Therefore, they can provide very valuable information about the formation of structures from high redshifts until today. This angular correlation function can be expressed as

$$C_{ij}^{\gamma\gamma}(\ell) = \frac{c}{H_0}\int \frac{\hat{W}_i^{\gamma(z)}\hat{W}_j^{\gamma}(z)}{E(z)r^2(z)}P_{\Phi+\Psi}(k_\ell,z)dz\,, \tag{181}$$

where $E(z) = H(z)/H_0$ is the dimensionless Hubble function, $\hat{W}_j^{\gamma(z)}$ are window functions, or lensing kernels, that project the redshift distributions and the power spectrum into angular scales, and finally, $P_{\Phi+\Psi}(k_\ell,z)$ is the Weyl power spectrum, which is related to the matter power spectrum $P_{\delta\delta}$ by

$$P_{\Phi+\Psi} = \Sigma^2(k,z)\left[3\left(\frac{H_0}{c}\right)^2\Omega_{\text{m}}^0(1+z)\right]^2 P_{\delta\delta}\,. \tag{182}$$

In this equation, we can see clearly the observational signature of the $\Sigma$ lensing function defined above in (154) and (157). We refer the reader again to [273] and the many references therein for details on the formulae of weak lensing. In Figure 10, we show the non-linear matter power spectrum $P(k,z)$ for $\Lambda$CDM (in light blue), K-mouflage (in green), JBD (in orange) and nDGP (in red) computed with privately modified versions of `MGCAMB, hi_class and EFTCAMB`. The models and their fiducial values have been chosen to be close enough to $\Lambda$CDM, to be still allowed by observations, but far enough so that distinctive changes can be measured with next-generation surveys. The standard cosmological parameters set for this specific prediction are $\Omega_{m,0} = 0.315$, $\Omega_{b,0} = 0.05$, $h = 0.674$, $ns = 0.966$ and $\sigma_8 = 0.8156$.

For JBD, the model parameter $\omega_{BD}$ is set to $\omega_{BD} = 800$, while for nDGP, the observational parameter is $\Omega_{rc} = c^2/(4r_5^2 H_0^2)$, where $r_5$ is the crossover scale defined above in (61) and we set here $\Omega_{rc} = 0.25$. For K-mouflage, the physical parameter is $\epsilon_2 = d \ln A(a)/d \ln a$, and it is related to the fifth force enacted by the scalar field, which comes from the conformal transformation of the metric (see [279] for more details). The prediction shown here is made for the case $\epsilon_2 = -0.04$.

These distinctive features can be observed when taking the ratio to $\Lambda$CDM for the three cosmological models considered above. While at linear scales $k \lesssim 0.07\,\mathrm{h/Mpc}$, the models show only a slight change in amplitude compared to $\Lambda$CDM (with nDGP showing the largest amplitude increase of about 10%), it is clear that for small scales there are distinctive features at play that dampen the power spectrum. In the right panels of Figure 10, we show the angular cosmic shear (weak lensing) power spectra in the 1,1 bin (lower redshifts) defined in (181) for all three screened models defined above. Furthermore, the ratio of the weak lensing $C_\ell$ with respect to $\Lambda$CDM is shown in the lower panel. In this case, the very sharp features observed in the matter power spectrum are smoothed out by the projection along the line of sight and into angular multipoles.

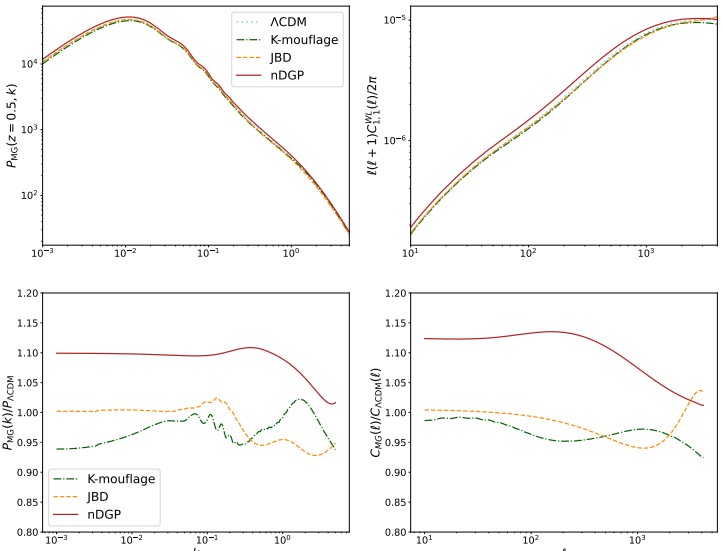

**Figure 10.** (**Upper left**): Matter power spectrum $P(k, z)$ for $\Lambda$CDM (in light blue), K-mouflage (in green), JBD (in orange) and nDGP (in red) computed with privately modified versions of `MGCAMB, hi_class and EFTCAMB`. (**Lower left**): Ratio of $\Lambda$CDM for the matter power spectra for the three cosmological models considered above. While at linear scales $k \lesssim 0.07\,\mathrm{h/Mpc}$, the models considered show only a slight change in amplitude compared to $\Lambda$CDM, at smaller scales there are some distinctive features, such as shifts in the BAO peaks and damping of power at small scales. (**Upper right**): Angular cosmic shear (weak lensing) power spectra for the 1,1 bin (lower redshifts) defined in (181) for the models mentioned above. (**Lower right**): Ratio of the weak lensing $C_\ell$ for screened modified gravity with respect to $\Lambda$CDM. The distinctive features observed in the matter power spectrum are smoothed out by the projection along the line of sight and into angular multipoles.

### 5.5. Going Beyond Linear Scales

At the linear level of perturbations in the matter density and the scalar field, these equations above can be computed very efficiently using modified versions of Einstein–Boltzmann codes, in particular of `CAMB`[16] (Code for Anisotropies in the Microwave Background) (see [280]), which is written mainly in `fortran`, and `CLASS`[17] (see [281,282]), which is mainly written in the `C` programming language. Both of these codes come with user-friendly `python` wrappers. The most common modifications of these codes accounting for theories of modified gravity and dark energy are based on two types; the first one is codes in which generic parametrisations of the deviations of GR as in (153) to (156) are

used. The second one is codes in which specific modified gravity (MG) models or a generic class of models are implemented and their full scalar field equations are solved, beyond the quasi-static approximation. From the first type, the two more common are `ISitGR`[18] (see [283,284], and `MGCAMB`[19] [285,286] and more recently a branch of `CLASS`, called `QSA_CLASS` (see [287]). For the second type, we will mention here the two most important ones, namely `hi_class`[20] (see [288] and `EFTCAMB`[21] (see [289,290]).

Up to now, we have only developed the formalism to compute the perturbations of matter and the field at the linear level. However, in order to study correctly and accurately the power spectrum and compare it with observations of galaxy clustering and weak lensing, one must go beyond linear scales. For galaxy clustering, the region around $k \approx 0.1 \mathrm{Mpc}^{-1}$, where Baryon Acoustic Oscillations (BAO) and redshift space distortions (RSD) are important, needs to be treated perturbatively in order to make accurate predictions. This involves using either Eulerian or Lagrangian perturbation theory [291,292] and furthermore using resummation techniques to capture accurately the large scale contributions [293,294]. For smaller scales, formalisms such as the effective field theory of large scale structures are needed in order to take into account the UV divergences of the perturbative models [295]. For the models we are interested in here, there has been some recent work by [235,296], some new work on GridSPT by [297] and some more foundational work by [298,299].

To obtain meaningful constraints with weak lensing, the power spectrum needs to be calculated at even higher $k$-values, for up to $k \approx 10 \mathrm{Mpc}^{-1}$, which is only possible using N-body simulations, which capture the full evolution of the non-linear gravitational dynamics. In scalar field models and especially in models that invoke screening mechanisms, these simulations are extremely computationally expensive and numerically complicated, since the non-linear evolution of the field needs to be taken into account. Several interesting approaches have been taken in the literature such as `COLA` [237], `Ramses` [300], `Ecosmog` [301], `MG-evolution` [302], $\phi$-`enics` (an interesting finite element method approach that can capture the non-linear evolution of the scalar field and reproduce very accurately the Vainshtein screening) [303] and the simulation work on $f(R)$ theories by several groups [304–307]. Since these simulations are time-consuming, faster approaches that allow for an efficient exploration of the parameter space would be extremely valuable and would be included in forecasts and Monte Carlo parameter estimation. Several approaches include fitting formulae based on simulations [308], emulators for $f(R)$ theories [309,310] and hybrid approaches in which the halo model, perturbation theory and simulations are calibrated to create a model, such as `REACT` (see [311,312]). This code can compute predictions for nDGP and $f(R)$ models, which are roughly 5% accurate at scales $k \lesssim 5\,\mathrm{h/Mpc}$.

### 5.6. Constraints on Screened Models with Current Data

In this section, we will focus on the constraints on different screened scalar-field models with current observations from CMB, background expansion and large scale structures.

### 5.6.1. Constraints on $f(R)$ Models

From the CMB, constraints have been placed by the Planck collaboration on the $f(R)$ model in terms of the Compton wavelength parameter, which is defined as

$$B \equiv \frac{f_{RR}}{f_R} R' \frac{H}{H'}, \qquad (183)$$

and its value today $B_0$ is related to the fundamental parameter $f_{R,0}$. Indeed, we have the relation

$$B = \frac{\Omega_m}{1+\omega} \frac{H^2}{m_{f_R}^2} \qquad (184)$$

where $\omega$ is the equation-of-state of the Universe and $m_{f_R}$ is the mass of the scalaron (166). Notice that the denominator $1+\omega$ is very small, and therefore, $B$ is less suppressed than the

ratio $H^2/m_{f_R}^2$. In the analysis of [313], the datasets used were Planck TT+lowP+BAO+SNIa + local H0 measurements (these last three observations are usually abbreviated as BSH), while CMB lensing was used to remove the degeneracy between $B_0$ and the optical depth $\tau$. At the 95% confidence level, they found $B_0 < 0.12$ with Planck data alone and when BAO, weak lensing (WL) and RSD were added, a much more stringent bound of $B_0 < 0.79 \times 10^{-4}$ was found, which forces the model to be very close to $\Lambda$CDM.

A very comprehensive, but by now relatively outdated, analysis by [314] using WMAP5 CMB data [315] and cross-correlations of ISW with galaxy clustering data provided interesting bounds on the variations of the gravitational potentials on an interesting redshift range $0.1 < z < 1.5$. For $f(R)$ models that follow the same expansion of the universe as $\Lambda$CDM they obtained a bound of $B_0 < 0.4$ at the 95% confidence level (CL). In the analysis by [316], large scale structure data coming from WiggleZ, BAO (from 6dF, SDSS DR7 and BOSS DR9, see [252–254]) were combined with Planck-2013 CMB [250] data and WMAP polarisation data [315] to find $\log_{10} B_0 < -4.07$ at the 95% CL. A more recent paper [317] uses the designer approach to $f(R)$ and tests it with Planck and BAO data. In this designer approach, one can fix the evolution of the background and then find the corresponding scalar field model that fits these constraints. With this, the bound of $B_0 < 0.006$ (95%CL) for the designer models with $w = -1$ is obtained, and a bound of $B_0 < 0.0045$ for models with varying equations-of-state is reached, which was then constrained to be $|w + 1| < 0.002$ (95%CL). All these bounds imply that $f(R)$ models cannot be self-accelerating, and also, if they are present, their background expansion will be very close to the one of $\Lambda$CDM according to observational bounds. This confirms the known results from gravitational tests in the solar system.

### 5.6.2. Constraints on nDGP Models

The self-accelerating branch of DGP (sDGP) has been plagued with the presence of ghost fields; nevertheless, it has been compared to observations, most recently in [318,319] where it was found, after using Planck temperature data, ISW and ISW-galaxy-cross-correlations, together with distance measurements that these models are much disfavoured compared to $\Lambda$CDM. The normal branch of DGP (nDPG) is non-self-accelerating, but it is still of interest since it shows clear deviations at scales important for structure formation. In [320], it was shown that the growth rate values estimated from the BOSS DR12 data [321] constrains the crossover scale $r_5$ of DGP gravity in the combination $[r_5 H_0]^{-1}$, which has to be $< 0.97$ at the $2\sigma$ level, which amounts to $r_5 > 3090$ Mpc/h, meaning that $r_5 \sim H_0^{-1}$, therefore making this model very similar to GR within our Hubble horizon. Further tests of this model against simulations and large scale structure data have been performed in [322,323].

### 5.6.3. Constraints on Brans–Dicke Theory

As mentioned previously, the most stringent constraint on JBD comes from solar system tests, where the Cassini mission put the bound of $\omega_{BD} > 40,000$ (see [33,324]). However, under an efficient screening mechanism (invoking a specific potential), the theory could still depart considerably from GR at cosmological scales. In an analysis by [325], the authors used Planck [250], WMAP [315], SPT and ACT [326,327] data plus constraints on BBN to set bounds on the JBD parameter. They assumed the scalar field to have initial conditions such that the gravitational constant would be the Newton constant today. With this, they found $\omega_{BD} > 692$ at 99% C.L. When the scalar was free and varied as a parameter, they found $\omega_{BD} > 890$, which amounts to $0.981 < G_{\text{eff}}/G_N < 1.285$ at the 99% C.L. In a more recent analysis by [261], the authors used the combined data of the Planck CMB temperature, polarisation, and lensing reconstruction, the Pantheon supernova distances, BOSS measurements of BAO, along with the joint $3 \times 2$pt dataset of cosmic shear, galaxy-galaxy lensing, and galaxy clustering from KiDS and 2dFLenS. They took into account perturbation theory and N-body calculations from `COLA` and `RAMSES` to compute the theoretical predictions for the power spectrum. They constrain the JBD

coupling constant to be $\omega_{BD} > 1540$ at the 95% C.L. and the effective gravitational constant, $G_{\text{eff}}/G = 0.997 \pm 0.029$. They also found that the uncertainty in the gravitational theory alleviates the tension between KiDS, 2dFLenS and Planck to below $1\sigma$ and the tension [242] in the Hubble constant between Planck and the local measurements to $3\sigma$. Despite these improvements, a careful model selection analysis shows no substantial preference for JBD gravity relative to $\Lambda$CDM.

### 5.6.4. Constraints on Horndeski Theories and Beyond

For Horndeski models, there has been a great effort by the Planck collaboration to test the parametrised deviations of GR such as in (174) and (175) or in the $\alpha$-formalism of [262]. However, in order to do so, certain conditions and restrictions on these parameters have to be met, given the relatively limited constraining power of current data. The code used in this case is the `EFTCAMB` code mentioned in Section 5.5.

In the Planck 2015 modified gravity paper [313], the authors considered Horndeski models with $\alpha_M = -\alpha_B$, $\alpha_T = \alpha_H = 0$, and $\alpha_K$ was fixed to a constant. This amounts to consider non-minimally coupled K-mouflage type models as in [262], with the only free function being $\alpha_M$. Additionally, the analysis used the ansatz,

$$\alpha_M = \alpha_M^{\text{today}} a^p \tag{185}$$

where $\alpha_M^{\text{today}}$ is a constant and $p > 0$ determines its backward time evolution. Furthermore, they relate the evolution of $\alpha_M$ to a linear ($p=1$) and exponential ($p > 1$, varying free) parametrisation [313]. Using the Planck TT+TE+EE+BSH data set combination (BSH standing again for BAO, SN and local Hubble constraints) they find $\alpha_M^{\text{today}} < 0.043$ (95% confidence level) for the linear case and $\alpha_M^{\text{today}} < 0.062$ and $p = 0.92_{0.24}^{0.53}$ (95% confidence level) for the exponential case. $\Lambda$CDM is recovered for $p = 1$ and $\alpha_M^{\text{today}} = 0$, therefore placing relatively strong limits on possible deviations of Einstein's GR.

As we discussed above, the gravitational wave event GW170817 constrained the Horndeski theory to be effectively composed only of Brans–Dicke models and cubic Galileons, and the latter are effectively ruled out by ISW observations. This then limits the interest on an overall analysis of Horndeski models in general. However, in [328], the authors analysed Horndeski models that still can have non-trivial modifications to GR, possible at the level of linear perturbations, and they confirmed the conjecture by [265] that $(\Sigma - 1)(\mu - 1) \geq 0$ for surviving models.

As an extension beyond this review, DHOST models, as mentioned above, can also provide an interesting phenomenology and are able to evade certain constraints affecting the Horndeski theories. References [329,330] studied DHOST models that present self-acceleration, and Reference [331], among others, have studied the astrophysical signatures of these models. However, their theoretical modelling has not been implemented yet in computational tools capable of analysing the full Planck CMB dataset. Finally, the authors of [332] performed a cosmological constraint analysis, assuming the form $\alpha_i = \alpha_{i,0} a^\kappa$ on these surviving Horndeski models, and using Planck and BICEP2/Keck [333] CMB data and galaxy clustering data from SDSS and BOSS, they found that when setting the kineticity to the following value $\alpha_K = 0.1a^3$, the $\alpha_{M,0}$ parameter has an upper limit of 0.38 when $\alpha_{B,0} \neq 0$ and 0.41 when $\alpha_{B,0} = 0$ at the 95% C.L. More importantly, they conclude that the effects of Horndeski theory on primordial B-modes (which at the time were expected to be measured accurately by BICEP/KECK2) are constrained by CMB and LSS data to be insignificant at the 95% C.L. However, they draw the attention to the fact that the assumptions on some parameters, for example, the assumed form of the kineticity, have major and dramatic effects on these results. In conclusion, the theory space of Horndeski models has been mostly ruled out by measurements of the ISW effect and the combination of CMB and large scale structure, when considering the gravitational wave event GW170817 and its electromagnetic counterpart GRB170817A. On the other hand, beyond Horndeski theories, such as DHOST, seem promising, but computational tools required to do a proper

cosmological analysis are not available yet, so the models can only be constrained by astrophysical observations so far.

## 6. Conclusions and Perspectives

Scalar-tensor theories are among the most generic and plausible extensions to $\Lambda$CDM, with potential relevance to much of astrophysics and cosmology. They must be screened to pass solar system tests of fifth forces. In this review, we have presented the most commonly screened modified gravity mechanisms and introduced them using an effective field theory point of view. The effective point of view is taken by first selecting a background, which could be cosmological, astrophysical or local, in the solar system. The coefficients of the different operators depend on the environment. This is a feature of all the screening mechanisms—physics is dependent on the distribution of matter—and gives them relevance to various different types of environments on a range of scales.

The screening mechanisms can be divided into two categories. The non-derivative mechanisms consist of the chameleon and Damour–Polyakov cases. The derivative ones are the K-mouflage and Vainshtein scenarios. The latter lead to scale-independent modifications of gravity on large scales. For models with derivative screening and having effects on large cosmological scales, the effects on smaller scales are reduced due to the strong reduction in fifth force effects inside the K-mouflage and Vainshtein radii. Nonetheless, the force laws on short scales in these scenarios deviates from $1/r^2$ and leads to effects such as the advance of the periastron of planets and effective violation of the strong equivalence principle in galaxies, both of which afford tight constraints. However, there is still some capability for ground-based experiments to test Vainshtein-screened theories [91]. The time dependence induced by the cosmological evolution is not screened in K-mouflage and Vainshtein screened models, which also leads to tight bounds coming from solar system tests of gravitation.

The chameleon and Damour–Polyakov mechanisms, on the other hand, have effects on scales all the way from the laboratory to the cosmos and must be taken on a case-by-case basis for each experimental setup and astrophysical observation. This makes the comparison between the short and large scale physics richer and leads to more complementarity between astrophysical and laboratory tests. For the symmetron, an experiment with a length scale between objects $d$ typically best constrains theories with mass parameter $\mu \approx d^{-1}$. If the mass were larger, then the scalar force between objects would be exponentially suppressed (as in (114)), while if it were smaller, the field would remain near $\phi = 0$, where it is effectively decoupled from matter. It is, therefore, desirable to employ a range of tests across as many length scales as possible. There is a notable exception to this general rule: if the ambient matter density between objects is of the order of the symmetry-breaking value $\rho_{\mathrm{amb}} \approx \mu^2 M^2$, then the symmetron is essentially massless. This enables even long-ranged experiments to test symmetron theories with $\mu \gg d^{-1}$ at that particular value of $M$.

The chameleon does not have a fixed mass parameter and hence there is more overlap between various experiments' capabilities to test the theory. Here, the differentiating feature tends to be when objects of a particular size become screened. If a given experiment's source and/or test mass is screened, then the experiment's capability to test the theory is strongly suppressed. Small values of the chameleon parameters $\{M, \Lambda\}$ correspond to even microscopic objects being screened, so only small-scale experiments are able to test that region of parameter space. One can observe this general trend in Figure 1: the bottom-left corner is constrained by particle physics experiments, the middle region by atomic-scale experiments, and the upper-right region by experiments employing macroscopic test masses such as a torsion balance. This trend continues with astrophysical tests constraining the region further above and to the right of the parameter space illustrated in the figure.

We have seen that, although screening mechanisms are easily classified, empirical testing is most often performed at the model level. Some of these models are archetypal, such as the $f(R)$ models of the Hu–Sawicki type for chameleons, the symmetrons for Damour–Polyakov, and the nDGP model for Vainshtein. For K-mouflage, there is no such

template, although specific models such as the "arctan" are promising because they pass the solar system tests. On cosmological scales, it is easier to test many theories at once, e.g., through the effective field theory of dark energy. Unfortunately, the link between the large scales and the small scales where screening must be taken into account is then lost. This is also a problem on cosmological scales where non-linear effects must be taken into account for weak lensing, for instance, and bridging the gap beyond perturbation theory and highly non-linear scales necessitates tools such as N-body simulations, which may be computationally expensive. A parameterisation of screening mechanisms valid from laboratory scales to the cosmological horizon would certainly be welcome. In the realm of non-derivative screenings, a parameterisation that exists and depends only on the mass and coupling dependence as a function of the scale factor the Universe allows to reconstruct the whole dynamics of the models, on all scales [53,88]. The same type of parameterisation exists for K-mouflage where the coupling function $A(a)$ and the screening factor $Z(a)$ are enough to reconstruct the whole dynamics too [279]. For Vainshtein and generalised cubic models defined by the function $G_3$, this should also be the case, although it has not yet been developed.

Fortunately, the space of theories that still need to be tested has drastically shrunk in the last few years. The models with the Vainshtein mechanisms and some influence on large scales are restricted to theories parameterised by one function $G_3$, which must be non-trivial because the simplest case, the cubic Galileon, has been excluded by observations of the Integrated Sachs Wolfe effect. Quartic and quintic Galileons are powerfully constrained by GW170817, the observation of a gravitational wave event with a near-simultaneous optical counterpart. Of course, theories with the Vainshtein property and no link with the cosmological physics of late-time acceleration of the Universe's expansion are fine, although the parameter space is restricted by galaxy-scale tests. On the thin-shell-screening side, wide regions of chameleon and symmetron parameter space are ruled out by laboratory probes and a largely complementary part by astrophysical tests involving stars and galaxies. The $n = 1$ Hu–Sawicki theory—the workhorse chameleon-screened model for over a decade—is now constrained by galaxy morphology to the level $f_{R0} < 1.4 \times 10^{-8}$ [185], such that it can no longer have appreciable astrophysical or cosmological effects. The phenomenological description of DHOST models is less developed, and it would be interesting to see whether and how these models could answer some of the pressing questions in cosmology such as the origin of the acceleration.

Future observations on cosmological scales from upcoming surveys such as Euclid will certainly provide a host of new results on screened models such as K-mouflage or nDGP. Only recently has it been realised that galactic scales afford strong probes of screening, and many more tests will likely be developed in the future. In the solar system, future satellite tests [334], which will test the equivalence principle down to a level of $10^{-17}$ in the Eötvos parameter, should also constrain screening mechanisms of the non-derivative type [335,336]. Finally, laboratory experiments ranging from the search for new interaction with Casimir configurations to atom interferometry should also provide new possibilities for the detection of screened modified gravity. While we have focused in this review on the relevance of screened scalar fields to the physics of dark energy, it may also be relevant to the other missing pillar of $\Lambda$CDM, dark matter. This is a key target for many upcoming astrophysical and cosmological surveys. Much less is known about screening in this regard, although fifth forces are clearly degenerate with dark matter in determining diverse objects' dynamics.

In conclusion, screening is a crucial ingredient in the physics of light scalar fields. Testing it with the next generation of experiments and observations may well lead to surprises and new discoveries.

**Author Contributions:** All the authors are responsible for the whole review. All authors have read and agreed to the published version of the manuscript.

**Funding:** P.B. acknowledges support from the European Union's Horizon 2020 research and innovation programme under the Marie Skodowska-Curie grant agreement No 860881-HIDDeN. H.D. is supported by a McWilliams Fellowship.

**Institutional Review Board Statement:** Not applicable.

**Informed Consent Statement:** Not applicable.

**Data Availability Statement:** Not applicable.

**Conflicts of Interest:** The authors declare no conflict of interest.

**Acknowledgments:** We thank Jeremy Sakstein for useful discussions. We thank Emilio Bellini, Noemi Frusciante, Francesco Pace, Ben Bose, Patrick Valageas and Giampaolo Benevento for providing private codes and input files to generate Figure 10.

## Notes

1  A term in $\varphi^2 \eta_{\mu\nu}$ could also be introduced leading to a contribution to the mass of the scalar field proportional to $\delta T$. This term represents a density-dependent contribution to the scalar mass, which would naturally occur in the case of the chameleon mechanism as the perturbation to the scalar mass by the local overdensity and does not alter the discussion that follows.

2  This follows from the coupling of the Newtonian potential to matter, $\mathcal{L}_N = -\Phi_N \delta T$.

3  One can also introduce the screening factor $\lambda_A = \frac{\beta_A}{\beta(\phi_0)}$ whereby screening occurs when $\lambda_A \leq 1$. The screening factor is also related to the mass of the thin shell $\Delta M_A$ as $\frac{\Delta M_A}{M_A} = 3\frac{\Delta R_A}{R_A} = \lambda_A$ where $\Delta R_A$ is its width and $(M_A, R_A)$ are respectively the mass and the typical radius of the object.

4  To be pronounced as camouflage.

5  Equation (21) should be understood as integrated over a ball of radius $r$. The left hand side is proportional to the point mass and the right hand side to the volume of the ball.

6  This inequality can be understood as $\Delta \Phi_N \geq \frac{\Lambda^5}{2\beta(\phi_0)m_{\mathrm{Pl}}} \int d^3 r \Delta^{-1}(r)$ where the integration volume is taken as a ball of radius $r$ and $\Delta^{-1}(r) = -\frac{1}{4\pi r}$.

7  As the background metric is the Minkowskian one, the use of Fourier modes is legitimate.

8  This theorem states that only potentials in $1/r$ and $r^2$ lead to closed trajectories.

9   The usual Casimir interaction due to photon fluctuations is obtained using Dirichlet boundary conditions for the electromagnetic modes corresponding to the limit of infinite fine structure constant [93]. In the scalar case, the same Dirichlet boundary conditions correspond to the limit where the density in the boundaries is considered to be very large compared to the one in the vacuum between the plates. In this case, the minimum of the effective potential almost vanishes in the plates. This applies to screening models of the chameleon or Damour–Polyakov types. For K-mouflage and Vainshtein screenings, the scalar profile is dictated by the presence of the Earth, and therefore, the plates have very little influence and thus do not lead to classical and quantum effects. The only exception to this rule appears for Galileon models where planar configurations do not feel the field induced by the Earth. In this case, planar Casimir experiments lead to a constraint on the conformal coupling strength $\beta \leq 0.05$ [91].

10  This reasoning, as we will see, does not apply to the symmetron case as the field vanishes between two plates when very light.

11  The response of scalar fields coupled to pointlike objects was considered in detail in [117,118], but for our purposes, the approximate result of Equation (113) will suffice.

12  https://www.desi.lbl.gov/ (accessed on 15 November 2021)

13  https://www.euclid-ec.org/ (accessed on 15 November 2021)

14  https://www.lsst.org/ (accessed on 15 November 2021)

15  https://roman.gsfc.nasa.gov/ (accessed on 15 November 2021)

16  https://camb.info (accessed on 15 November 2021)

17  https://class-code.net (accessed on 15 November 2021)

18  https://labs.utdallas.edu/mishak/isitgr/ (accessed on 15 November 2021)

19  https://github.com/sfu-cosmo/MGCAMB (accessed on 15 November 2021)

20  http://miguelzuma.github.io/hi_class_public/ (accessed on 15 November 2021)

21  http://eftcamb.org (accessed on 15 November 2021)

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
