# Peer review of "Testing Screened Modified Gravity"

_universe, doi:10.3390/universe8010011_

Round 1
Reviewer 1 Report
The paper "Testing Screened Modified Gravity" by authors Brax, Casa, Desmond and Elder considers screening mechanisms in the context of modified/extended gravity. These mechanisms constitute an essential, and commonly natural, feature for almost any model of gravity that deviates from general relativity at astrophysical or cosmological scales.
The authors have already contributed with several important papers on the subject, they are clearly leading researches in the field covered by this review.
The text is very clear and it can work as an excellent and up-to-date introduction to the subject.
Apart from small issues, as detailed below, I fully agree with the publication of this review in Universe journal.
I would like to point out a few suggestions to improve further the clarity. They are listed below:
* Eq. (1) is clearly a very important equation since it puts out the framework of the screening within the effective field approach. However, its motivation at this point in the text I found obscure. As far as I understand, anything beyond the contribution of the two first terms of eq. (1) (including contributions that are not commonly identified with the matter energy momentum tensor), should be incorporated in the \delta g \delta T term. A brief explanation about the latter and the conservation of \delta T with respect to the background metric would improve clarity.
* Eq. (2), since the term \partial \phi \partial \phi was considered, is there any reason for not considering a term of the type \eta_{\mu \nu} \phi^2 ?
* Eqs. (13) and (14), shouldn't there be an absolute value on \Phi_N as well? This since \Phi_N should be negative for bound systems.
* Eq. (15): I have the impression that the true condition should be Z >> \beta, not Z >> 1.
* Below eq. (16) there is a reference to M, but there is no M in the above eq., probably it is a reference to m_Pl.
* Below eq. (27) it is stated that chameleons correspond to k = 0, implying that screening depends on the value of the Newtonian potential alone. However, going back to eq. (13), one sees that the condition depends on the ratio \varphi/\Phi_N. For small mass \varphi \propto \Phi_N, hence there is no dependence on \Phi_N. It would be nice to clarify this statement.
* Eq. (63), there seems to be a missing tilde in the first \sqrt{-g}.
* Above eq. (64), and considering eq. (63), I understand it should be Einstein frame metric \tilde g, not Jordan frame.
* Above eq. (134), perhaps "In the Newtonian limit of GR this is given by" is a better statement.
* About gravitational waves, their speed and Vainshtein screening, perhaps this PRL https://arxiv.org/abs/1507.05047 should be briefly commented.
* eq. (2), a \phi appears as if it were an index.
* Below eq. (6), there is a missing parenthesis.
* line 170, it is written "must a scalar-tensor"
* Above eq. (89), missing citation.
* There are several figure citations with a question mark. See for instance just below eq. (120), below eq. (182), line 986, line 1051...
* Part of Figure 1 is outside of the page.
* Above eq. (160): missing citation.
* Eq. (162): the parenthesis should be enlarged.
* Eq. (180): there is an undefined P_{ffiffi}. It also appears just above eq. (182).
* In the references, the arxiv links are not working. Also, there are several lower case "prd".
Author Response
Dear Editor,
WE would like to thank the referee for his careful reading of our manuscript. We have complied with the referee's requests and amended our manuscript accordingly. Here is a list of our changes:
1) We have added a sentence after eq.1 to specify that the composite metric is the perturbed Jordan metric and the energy momentum tensor conserved in this frame.
2) We have added a footnote. Indeed adding a $\varphi^2$ term is allowed but would only add a $\delta T$ perturbation to the mass which corresponds to adding the matter perturbation in the mass term, and therefore does not change the nature of the discussion concerning the chameleon screening mechanism.
3) We have added the absolute values of the Newtonian potential.
4) We have corrected a typo an screening is now correctly defined as the reduction of the effective coupling implying that $Z\gg 1$ should be required.
5) We have added a paragraph explaining that for very dense objects in a fixed environment, the chameleon screening criterion becomes $\vert \Phi_N\vert > C$.
6) In (63) and (64), the metric $g_{\mu\nu}$ is the Einstein metric in the Einstein-Hilbert term, and $\tilde g_{\mu\nu}$ is the Jordan metric appearing only in the matter action. It seems to us that there is no typographic error in these equations.
7) We have added a reference to Jimenez et al. as the paper was clearly the first one to worry about the speed of gravitational waves for these models.
8) We have corrected all the typos mentioned.
9) We apologise for some of the typos as they were due to the transfer by the publishing company between our prdtex original submission and the Journal's template. We hope that these problems are/will be sorted prior to publication (for instance the link with the arxiv).
We hope that our review is now in publishable form
Yours sincerely
The authors
Reviewer 2 Report
This article is a good review of Modified Gravity, state-of-the-art, subject in Cosmology written by experts on the field. This work is timely due to the upcoming generation IV experiments both in galaxy surveys and CMB measurements. The manuscript is well written and interesting. For these reasons I recommend its publication in Universe journal.
Author Response
Dear Editor,
We would like to thank the referee for his/her comments
Yours sincerely
The authors
Reviewer 3 Report
It is known that bounds on gravitation in the solar system and the laboratory are violated by long range scalar fields with a coupling to matter. Indeed, however, such an effect can be evated by the screening mechanisms.
In this review article, the various screening mechanisms are explained from an effective field theory point of view. Furthermore, it is explored how they can and will be tested in the laboratory and on astrophysical and cosmological scales.
The discussions are interesting and the manuscript is well written. Moreover, the detailed descriptions and explanations are given. Thus, this review article can be accepted for publication in Universe. Before publication, if it is possible, it is recommended that the following point should be considered.
There exist past related reviews on modified gravity theories including the screening effects. By comparing with these preceding works, the new ingredients and the strength of this work should be stated more explicitly and in more detail.
Author Response
Dear Editor
we would like to thank the referee for his/her comments. We have complied with the requested changes and have added a long paragraph at the end of the introduction where we compare our work to the existing literature.
Yours sincerely
The authors.